

# The XSO framework (v0.1) and Phydra library (v0.1) for a flexible, reproducible and integrated plankton community modeling environment in Python

Benjamin Post[1,2], Esteban Acevedo-Trejos[3], Andrew D. Barton[4], and Agostino Merico[1,2]

[1]Systems Ecology Group, Leibniz Centre for Tropical Marine Research (ZMT), Bremen, Germany
[2]School of Science, Constructor University, Bremen, Germany
[3]Earth Surface Process Modelling, GFZ German Research Centre for Geosciences, Potsdam, Germany
[4]Scripps Institution of Oceanography and Department of Ecology, Behavior and Evolution, University of California San Diego, La Jolla, CA, United States

**Correspondence:** Benjamin Post (benjaminpost@aoop.de)

**Abstract.**

Plankton community modeling is a critical tool for understanding the processes that shape marine ecosystems and their impacts on global biogeochemical cycles. These models can be of variable ecological, physiological and physical complexity. Many published models are either not publicly available or implemented in monolithic and inflexible code, thus hampering adoption, collaboration, and reproducibility of results. Here we present *Phydra*, an open-source library for plankton community modelling, and *Xarray-simlab-ODE (XSO)*, a modular framework for efficient, flexible, and reproducible model development based on ordinary differential equations. Both tools are written in Python. Phydra provides pre-built models and model components that can be modified and assembled to develop plankton community models of various levels of ecological complexity. The components can be created, adapted and modified using standard variable types provided by the XSO framework. XSO is embedded in the Python scientific ecosystem and is integrated with tools for data analysis and visualization. To demonstrate the range of applicability and how Phydra and XSO can be used to develop and execute models, we present three applications: (1) a highly simplified nutrient-phytoplankton (NP) model in a chemostat setting, (2) a nutrient-phytoplankton-zooplankton-detritus (NPZD) model in a zero-dimensional pelagic ocean setting, and (3) a size-structured plankton community model that resolves 50 phytoplankton and 50 zooplankton size classes with functional traits determined by allometric relationships. The applications presented here are available as interactive Jupyter notebooks and can be used by the scientific community to build, modify, and run plankton community models based on differential equations for a diverse range of scientific pursuits.

## 1 Introduction

Scientists have used mathematical models to advance our understanding of marine ecosystems for at least 70 years (Sverdrup, 1953; Fasham et al., 1990; Gentleman, 2002; Follows et al., 2007; Acevedo-Trejos et al., 2016). Early models comprising a few differential equations describing phytoplankton populations in a simplified physical setting (Evans and Parslow, 1985; Fasham et al., 1990) have matured into detailed descriptions of multiple trophic levels that are run in complex three-dimensional general



circulation models (GCMs) (e.g. Dutkiewicz et al., 2020). While plankton community models often lack biological realism and suffer from poorly-constrained model parameters and comparisons to observations (Anderson, 2005), they have been important in developing our understanding of the mechanisms shaping plankton biogeography (e.g. Follows et al., 2007), phenology (e.g.

Taylor et al., 1993), and biodiversity (e.g. Barton et al., 2010; Acevedo-Trejos et al., 2015), as well as links between ecosystems and biogeochemical cycles (e.g. Fasham et al., 1990; Sarmiento et al., 1998; Merico et al., 2006; Dutkiewicz et al., 2009).

Despite this progress, we argue that the technical implementations of plankton community models are often inflexible, complicated, and inaccessible, which obscures valuable research and presents a high barrier of entry for beginners or students. Existing model code is rarely reused beyond the development teams (Belete et al., 2017). Many models use legacy codes

that are difficult to modify or integrate, resulting in "good knowledge bound in outdated code" (Argent, 2004). This creates challenges, particularly when attempting to integrate models across domains, e.g., linking ecological models to sophisticated physical models (Koralewski et al., 2019) or when calibrating models (Steenbeek et al., 2021).

A collective and dedicated effort in the marine ecosystem modeling community is ongoing to improve on these issues. It has become more common to publish model source code, and there is an ongoing development of open-source frameworks, that

can make models more approachable, flexible, and reproducible (Janssen et al., 2015). On one end, there are large scale global models, often written in the highly efficient programming languages Fortran, that are systematically embedded in frameworks. Examples are the modular biogeochemical modeling suite MARBL (Long et al., 2021), or the limnological FABM-PCLake model (Hu et al., 2016). In these projects, generally, a large monolithic model code is modularized and partially retrofitted with a user interface, for example by allowing the user to supply a markup language file to initialize the model. Much of the

model is still hard-coded in the underlying Fortran scripts, such that advanced technical knowledge is necessary for granular control of model structure. Not all students entering the field of plankton community modeling will start working on such large ecosystem models, and Fortran is not among the first programming languages learned by beginners. Instead, students usually start with interpreted programming languages commonly used for data analysis applications, such as Python. These languages are typically designed with the aim of improving code structure and readability and have evolved the capabilities to efficiently

support advanced numerical computations (Lin, 2012), in part by wrapping lower-level languages such as Fortran or C++. This is showcased by Veros, a global circulation model (GCM) translated to Python (Häfner et al., 2018). The Python scientific ecosystem and Jupyter Notebooks in particular (Kluyver et al., 2016) have proven to be a useful tool for collaborative model development workflows (e.g. eWaterCycle platform, Hut et al., 2022).

To efficiently test and answer ecological and biogeochemical questions using plankton community models, we need model-

ing tools that: (1) are easy to use, (2) are open-source, (3) allow flexible and granular control of model structure, and (4) are conducive to scientific collaboration via an open and extensible framework. These motivations lead us to develop the novel *XSO* framework and *Phydra* library in the programming language Python. The XSO framework offers a set of building blocks for developing computational models based on ordinary differential equations. XSO is used as the basis of the plankton community models contained in the Phydra library. The foundational framework facilitates the modification of model structure,

dimensionality, and parameterization. The ultimate goal is to provide usability and flexibility in line with popular Python data



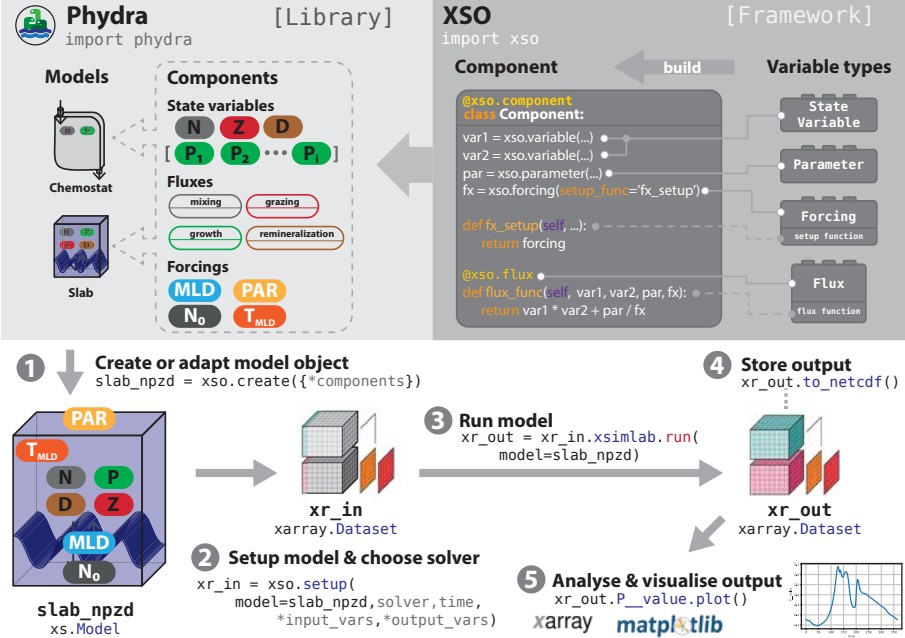

**Figure 1.** Schematic of a typical workflow utilizing XSO and Phydra. XSO provides the framework. Phydra is a library of functional *components* and pre-built *model objects*, that can be used, extended, and modified. A typical workflow would consist of five steps. (1) Choose a pre-existing model, potentially remove or add *components* or create a new model using `xso.create()`. (2) Create a *model setup* by supplying the appropriate labels, parameters and solver to `xso.setup()`. The *model setup* is an Xarray dataset. (3) To run the model, call the *xsimlab.run()* method on the *model setup*. Output is returned as a Xarray dataset containing all metadata. (4) These datasets can be easily stored or shared. (5) Xarray datasets are fully compatible for being analyzed and visualized with the wealth of tools provided by the Python scientific ecosystem.

analysis and visualization tools, such as Pandas, Xarray and Matplotlib. The XSO framework depends on functionality from these packages and provides direct interoperability for an integrated modeling environment.

In the next sections, we present the XSO framework and structure of the Phydra library, including the steps of an exemplary model development workflow. We show the utility of the tool-set in three exemplary model applications: (1) a basic nutrient-phytoplankton (NP) chemostat model, (2) a nutrient-phytoplankton-zooplankton-detritus (NPZD) model in a slab ocean physical setting, adapted from Anderson et al. (2015), and (3) a complex size-resolved plankton community model in a simple box setting, adapted from Banas (2011). These models form the basis of the first release of the Phydra library. We then discuss the architecture of the framework, current limitations, and possible future developments.





## 2    Descriptions of the XSO framework and the Phydra Library

### 65    2.1    The XSO framework

Xarray-simlab-ODE (XSO) is a Python framework that allows users to construct and customize models based on ordinary differential equations (ODEs) in a modular fashion. It is a non-opinionated framework, i.e., it does not provide a fixed notion of how a model should be implemented, instead it attempts to remove the redundant boilerplate code, allowing a user to efficiently construct and work with ODE-based models. XSO was developed as the technical foundation of the Phydra library,

but is not limited to any particular domain and can be used to create ODE-based models of any type. The typical steps of a model development workflow are presented in Fig. 1.

The XSO framework is an extension of Xarray-simlab (Bovy and Braun, 2018; Bovy et al., 2021), which itself provides a generic and highly flexible model development framework in Python. It relies on object-oriented Python functionalities, such as compact data classes and decorators (see the online-documentation for more details). Xarray-simlab provides a succinct

set of functions and attributes to construct Python objects, that can interact as processes of a larger model. In addition to this interface, Xarray-simlab provides powerful data handling capabilities, storing model input and output as multidimensional Xarray datasets (Hoyer and Hamman, 2017) including all relevant metadata (such as units of variables). Model output is thus directly compatible with a wealth of other Python tools for data analysis or visualization, and can be readily exported to the NetCDF file standard (amongst others).

Xarray-simlab has found various applications, for example in landscape evolution (Bovy, 2021) and plant growth modeling (Vaillant et al., 2022). The Xarray-simlab framework is generic in that it provides only a step-wise execution of model processes and could be utilized to build almost any kind of computational model. Our package XSO is, technically, a wrapper around Xarray-simlab, adding custom building-blocks and backend code to allow a user to easily define and compute models based on differential equations.

Our objective in developing the XSO framework was to enable users to construct ODE-based models to be readily modified, especially in relation to dimensionality and number of state variables and processes involved. XSO provides an interface for iterative modifications, both to more complex and simpler model constructs. The building blocks provided by XSO are as follows:

– *Variable types*: These are the most granular elements of the framework, which directly correspond to the basic mathemat-

ical components of ODE-based models (e.g., state variables, parameters, forcing, and partial equations). XSO currently provides the following *variable types*:

– `xso.variable`: Defines a state variable in a *component*, either locally or via reference in another *component*.

– `xso.forcing`: Defines an external forcing as a constant or time-varying value, via an additional setup function. Can also be a reference to a forcing in another *component*.

– `xso.parameter`: Defines a constant model parameter.





- – `xso.flux`: Defines a partial equation with the *variable types* within the *component*, and adds the term to the system of differential equations of the underlying model. The flux function decorator provides a `group` argument, that allows passing fluxes as arguments between components.

- – `xso.index`: Creates an input variable to define a dimension label (i.e., Xarray index) within the model, stored as metadata in the input and output dataset.

These can be used to define variables in compact Python classes, to construct functional XSO *components*. All of them can be defined with a variable number of dimensions (i.e., as a vector, array, or matrix).

- **Components**: These are the building-blocks of a model. *Components* declare a subset of variables and define a specific set of mathematical functions computed for these variables during model runtime. More specifically, a *component* refers to a Python class containing *variable types* that is decorated with the `@xso.component` function. For example, a *component* could define a specific nutrient uptake function, e.g. Monod-type phytoplankton growth on a single nutrient. The decorating function registers the *variable types* within the framework, reducing boilerplate code and creating fully functional model building blocks. *Components* can be reused within a model.

- **Model object**: These are instances of the Model class provided by Xarray-simlab. They consist of an ordered, immutable collection of *components*. A XSO *model object* is created with a call to the function `xso.create()` by supplying a dictionary of model *components* with their respective labels. *Model objects* contain the *components* relevant to a model and can be easily stored and shared. They do not contain custom parameterization.

- **Model setup**: This object is a Xarray dataset, that includes all relevant information needed at runtime, such as the *model object*, solver algorithm to be used, as well as time steps and model parameterization. A XSO *model setup* is created with a call to the function `xso.setup()` and supplying the aforementioned information as arguments. At this step, the *variable types* initialized in a *component* must be supplied with a value, as well as a label that can be used to reference them in other *components*. The model parameterization is passed as a dictionary, referencing the *component* labels and variable names.

The system of differential equations is constructed from the *fluxes* using the labels supplied during model setup. The number of values in a defined dimension is flexible, but they have to match across the model in order for the model to run. When executing the model by calling the `xsimlab.run()` method of the *model setup* dataset and supplying the appropriate *model object*, a "filled-out" Xarray dataset is returned containing model setup parameters, metadata, and output.

The XSO framework currently provides two solver algorithms: an adaptive step-size solver from the SciPy package `solve_ivp` (Virtanen et al., 2020) and a simple step-wise solver that is built into the backend Xarray-simlab framework. The `solve_ivp` algorithm is implemented to use the default `RK45` method, which is an explicit Runge-Kutta method of order 5(4) (Dormand and Prince, 1980). Apart from the technical limitations of the solver algorithm used, there are no restrictions to the dimensionality and number of *variable types* used within a *component* and no limitations to the levels of *group* variables linking *components* to define a single ecosystem process. The `xso` Python package is available via PyPI and Github (Post, 2023b).



## 2.2 The Phydra library

Phydra is a Python package that provides a library of modular plankton community models built using the XSO framework. Phydra establishes conventions and common usage for building models using XSO.

The plankton community models included in the Phydra package are available to the user at multiple hierarchical levels: as a library of pre-built XSO model *components*, as pre-assembled *model objects*, and as exemplary model simulations in interactive Jupyter notebooks. These levels are described below.

1. ***Components***: The first version of the library will contain all *components* used to create the three model applications presented in Section 3. The *components* can be combined to zero-dimensional plankton community models of variable complexity. The library follows common usage patterns and conventions. As long as the labelled model dimensions between *components* match at model setup, all *components* included in the Phydra library are compatible.

2. ***Model objects***: The first release of Phydra contains the *model objects* defined in the three model applications presented
in section 3. The *model objects* can be imported from the library and can be readily setup, modified, and run by a user.

3. ***Example notebooks***: *Model objects* only define the collection of *components*. To run a model, the input parameters still need to be defined and supplied at runtime. The Phydra library comes with three fully documented model applications that are presented in interactive Jupyter notebooks. These notebooks show all steps from creating the *model setup* object to analyzing model output and provide a template for further exploration and experimentation with the provided plankton
community models.

The open-source and extensible nature of Phydra and XSO enables users to customize and develop processes that accurately describe a particular ecosystem. In a collaborative effort aiming to promote efficient, transparent, and reproducible marine ecosystem modeling, Phydra encourages users to contribute their own *components* and *models* to the core library. The Phydra library could potentially offer a comprehensive, well-documented, and peer-reviewed codebase for the scientific exploration of
plankton community models. Phydra is available via Github (Post, 2023a).

## 3 Model applications

To showcase the utility of the XSO framework and Phydra library, we present three plankton community model applications of varying complexity. For each application, we present the mathematical model, the implementation within the XSO framework and the model results. To highlight the flexible nature of the model implementations, we also show how one aspect of each
model can be modified.

For the first application, we consider a simple chemostat model, whose implementation using the XSO framework is presented in full detail. For the presentation of the more complex models, we show only the *component* structure and highlight additional technical aspects of the implementation. For all use cases, the complete codes, following the full development work-



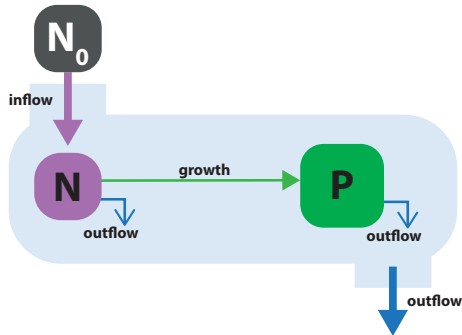

**Figure 2.** Schematic of model application 1: Phytoplankton $P$ consuming a single nutrient $N$ in a flow-through chemostat system. The chemostat system is supplied with external medium with nutrient concentration $N_0$. The medium flows into the system and both $N$ and $P$ flow out of the system at a constant rate, $d$.

flow from model creation to output visualization, are available publicly as interactive Jupyter notebooks in the "notebooks"
folder of the Phydra repository (Post, 2023a).

### 3.1 Model application 1: phytoplankton growth in a chemostat

Chemostats are a commonly used experimental setup for studying the growth dynamics of microorganisms under controlled laboratory settings. They are characterized by a constant inflow of the medium containing nutrients and a constant outflow of the culture, both at a fixed rate $d$ ($\mathrm{d}^{-1}$). Under constant conditions, a steady-state emerges that is particularly useful for
studying growth rates of microorganisms. Although the conditions of chemostat systems do not have a direct equivalent in nature, some oceanic upwelling systems can be approximated with such a simple model (Haefner, 2005).

To showcase the flexibility and simplicity of the XSO framework, we consider two cases: (1) a constant nutrient input and (2) a sinusoidal nutrient input (time-varying $d$).

#### 3.1.1 Description

The chemostat model is presented in Fig. 2. It comprises two state variables, dissolved nutrients ($N$) and phytoplankton ($P$). The model expresses quantities in units of µM N (i.e. $\mu\mathrm{mol\,N\,m^{-3}}$). The physical environment is a flow-through system corresponding to a laboratory chemostat setup. Growth medium with nutrient concentration $N_0$ (µM N) flows into the system at a rate $d$ ($\mathrm{d}^{-1}$). The model components ($N$ & $P$) flow out of the system at that same rate.

Phytoplankton growth $\mu$ ($\mathrm{d}^{-1}$) is described by Monod kinetics (Monod, 1942).

$$\mu = \mu_{max} \left( \frac{N}{k_N + N} \right) \tag{1}$$




**Table 1.** List of variables and parameters considered for the NP chemostat model. In addition to values and units, we report the variable names to compare with Fig. 3

| Description | Symbol | Variable | Value | Units |
|---|---|---|---|---|
| Nitrogen concentration | $N$ | N | t(0) = 1 | µM N |
| Phytoplankton concentration | $P$ | P | t(0) = 0.1 | µM N |
| External nitrogen concentration | $N_0$ | N_0 | 0.1 | µM N |
| Maximum growth rate | $\mu_{max}$ | mu_max | 1 | $d^{-1}$ |
| Dilution rate | $d$ | rate | 0.1 | $d^{-1}$ |
| Half-saturation constant | $k_N$ | halfsat | 0.7 | µM N |
| Sinusoidal mean | $m$ | mean | 1 | $µM\,N\,d^{-1}$ |
| Sinusoidal period | $p$ | period | 24 | d |
| Sinusoidal amplitude | $a$ | amplitude | 0.5 | $µM\,N\,d^{-1}$ |

where $k_N$ (µM N) is the half-saturation nutrient concentration, defined as the concentration at which half the maximum growth rate is achieved, $N$ is the ambient nutrient concentration, and $\mu_{max}$ ($d^{-1}$) is the maximum growth rate achievable under ideal growth conditions.

The model equations are:

$$\frac{dN}{dt} = d\left(N_0 - N\right) - \mu_{max}\left(\frac{N}{k_N + N}\right)P \tag{2}$$

$$\frac{dP}{dt} = \mu_{max}\left(\frac{N}{k_N + N}\right)P - dP \tag{3}$$

### 3.1.2 Implementation

To meaningfully structure our model within the XSO framework, we separate the model into state variables, forcing, and fluxes. For state variables, we have nutrient ($N$, Equation 2) and phytoplankton ($P$, Equation 3). The only forcing is the external nutrient concentration ($N_0$, Equation 2). Three fluxes can be defined: (A) the inflow of the external medium (Equation 2), (B) $P$ growing on $N$ (Equation 1, 2 and 3), and (C) the outflow of both $N$ and $P$ (Equation 2 and 3). The model is implemented using these 6 separate model components, as shown in Fig. 3.

To explore the basic model dynamics, we choose standard parameter values (Table 1). Initial values for $N$ and $P$ are set at 1 µM N and 0.1 µM N, respectively. The model is run for 100 days with a time step of 0.1 days.

In order to run the model with periodic forcing, we simply exchange the forcing component from `ConstantForcing` to `SinusoidalForcing` (see Fig. 3). This specific component requires two more input parameters, but otherwise the model creation and setup remain the same. We can update the model object, by simply exchanging the `SinusoidalForcing` component for the `"N_inflow"` component via the `model.update_processes()` method and updating the corresponding





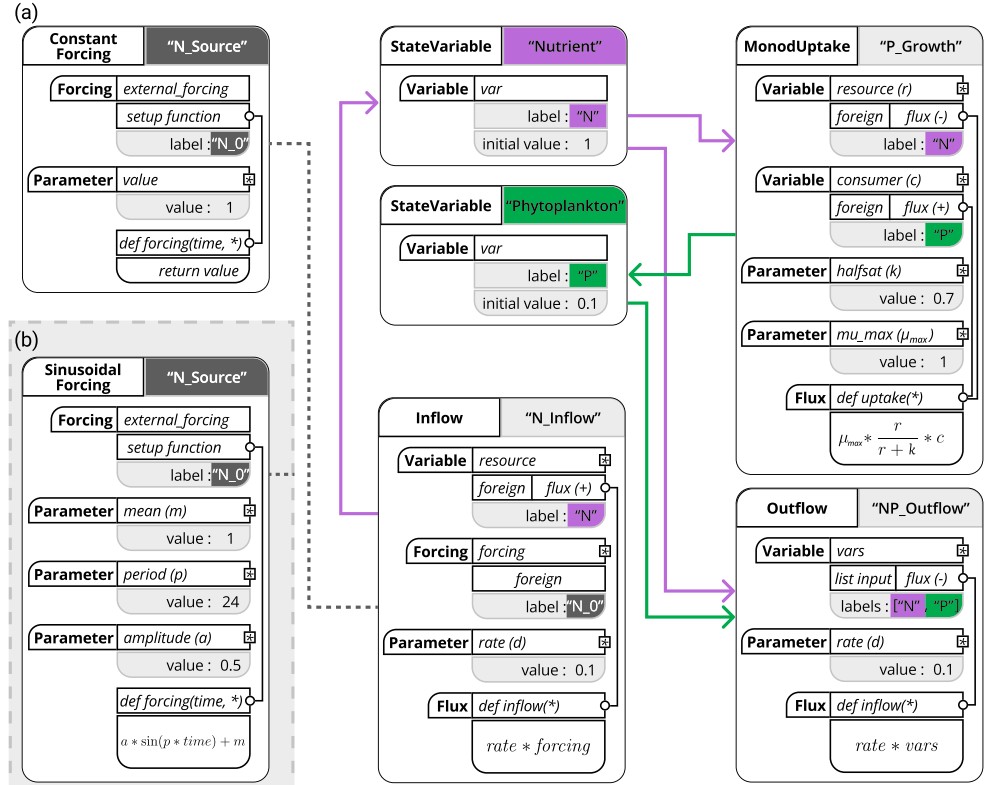

**Figure 3.** Schematic representation of the chemostat model using the XSO framework and included in the Phydra library. Model setup with constant forcing (a) and with sinusoidal forcing (b). Structures in solid black are hard-coded into components. Labels of the different components are supplied at model creation. Gray boxes and the resulting links between components (shown as thick colored arrows and dashed lines) are defined at model setup, via the supplied labels and parameters. The asterisks in the flux function input arguments references the variables, forcing and parameters defined within the same component, these local variables can be used in all functions (e.g. fluxes or forcing setup functions) within that same component.

parameters via the `model_setup.update_vars()` functions supplied by the Xarray-Simlab framework that XSO extends. Such functionality allows straightforward modification and testing of model structures.

### 3.1.3 Results

Fig. 4 shows the results of two cases considered. Under constant forcing, the model quickly reaches a steady state, as nutrient supply and the resulting phytoplankton growth balances with the loss of nutrient and phytoplankton due to the constant outflow. The periodically variable forcing creates oscillations in $P$ centered around $0.9\,\mu\mathrm{M\,N}$. In this highly simplified model, the results show the typical time shift between nutrient and phytoplankton, i.e. the time lag between the point in time when all nutrients in the culture are consumed and the peak in phytoplankton concentration.





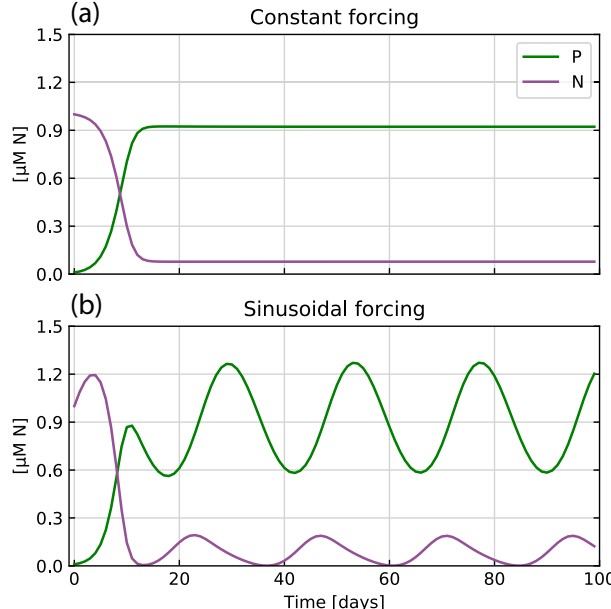

**Figure 4.** Model outputs for the two chemostat scenarios: (a) Constant forcing and (b) Sinusoidal forcing. In both cases, the concentration of nutrient ($N$, purple) and phytoplankton ($P$, green) are shown through time.

By producing expected results with a very simple model setup, this first application represents a basic proof-of-concept of our framework and library.

### 3.2 Model application 2: Nutrient-Phytoplankton-Zooplankton-Detritus (NPZD) model

The classic Nutrient-Phytoplankton-Zooplankton-Detritus (NPZD) model is embedded in a slab-ocean physical setting (e.g., Evans and Parslow, 1985; Fasham et al., 1990). "Slab" refers to a simplified zero-dimensional model of the oceanic upper mixed layer, which depth varies seasonally. This model structure provides an efficient physical setting for more complicated ecosystem descriptions and is used for both research and teaching purposes. This application is adapted from the EMPOWER model, as presented by Anderson et al. (2015). See Fig. 5 for a schematic of the model structure.

In the model, phytoplankton growth is driven by temperature, light, and nutrients. Phytoplankton are consumed by zooplankton, which are in turn subject to a higher order mortality (such as predation by higher trophic levels). Phytoplankton and zooplankton mortality and grazing by-products fuel a detrital pool that is remineralized in the upper ocean. Changes in the depth of the upper mixed layer have effects on all components. Nutrients are exchanged between the upper ocean and deep ocean across the mixed layer boundary. Fractions of phytoplankton, zooplankton, and detritus are lost due to mixing, with 215   detritus additionally sinking out of the mixed layer at a constant rate.

  Many NPZD-type models have been published over the years, with a variety of formulations for the functional responses of the ecosystem components. Anderson et al. (2015) showcase multiple alternative formulations, particularly focusing on the



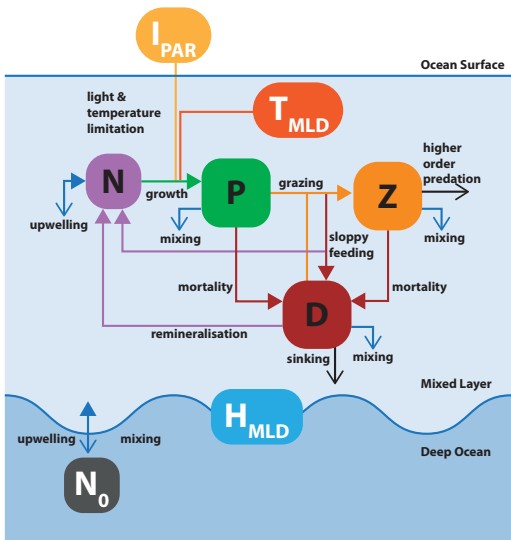

**Figure 5.** Schematic of model application 2: the Nutrient-Phytoplankton-Zooplankton-Detritus (NPZD) slab model. The model structure is adapted from Anderson et al. (2015). Boxes with black and white labels represent, respectively, state variables and external forcing. Arrows indicate fluxes between state variables. Filled colored arrows represent exchanges between state variables and forcings, open arrows represent fluxes that are lost from the model system. The upper layer box contains the ecosystem model, with state variables for nutrient, phytoplankton, zooplankton, and detritus. The oscillating blue line represents the seasonally variable mixed layer depth (MLD) that defines the boundary between the upper layer and the abiotic deep ocean.

treatment of light in the model. We partially follow their analysis by considering two different light-attenuation algorithms in our modular implementation with the XSO Framework.

### 3.2.1 Description

The model expresses quantities in units of μM N, with state variables for dissolved nutrients ($N$), phytoplankton ($P$), zooplankton ($Z$) and detritus ($D$). The water column is represented by two vertically stacked layers. One is the upper layer, containing the ecosystem, and the other is a biologically inert deep box. All symbols, parameter values and units are reported in Table 2. For a more detailed presentation of model structure and formulation, we refer the reader to the original publication (Anderson et al., 2015).

The model is driven by external forcing describing the depth of the upper mixed layer $H$ (m), the average temperature of the upper mixed layer $T$ (°C), photosynthetically active radiation (PAR) at the ocean surface $I$ (Wm$^{-2}$), and nutrient concentration in the deep layer $N_0$ (μM N).





The deeper layer supplies nutrients to the upper layer. Fractions of all state variables are lost into the deeper layer due to
mixing. The rate of mixing is described by $K$ ($\mathrm{d}^{-1}$):

$$K = \frac{h^+ + \kappa}{H} \tag{4}$$

Where $\kappa$ ($\mathrm{m\,d}^{-1}$) represents constant diffusive mixing. Variable mixing is a function of the change in mixed layer depth
(MLD) over time $h = \frac{dH}{dt}$. The function $h^+$ ($\mathrm{m\,d}^{-1}$) defines the differential effects of entrainment and detrainment due to the
changes in MLD as $h^+ = \max(0,\ h)$. When the mixed layer shallows, $h^+$ does not modify $K$, based on the assumption that
detrainment of mass and the increase in concentration due to the reduced volume of the mixed layer are balanced (Evans and
Parslow, 1985). We note that, for comparability, we follow the EMPOWER model in their treatment of motile entities ($Z$) as
having the same mixing term as non-motile entities ($N$, $P$, and $D$) (Anderson et al., 2015). Traditionally, motile entities are
treated differently (e.g., see Fasham et al. (1990)).

Dissolved nutrients in the mixed layer ($N$, $\mathrm{\mu M\,N}$) are supplied via mixing, the fraction of zooplankton excretion, and
remineralization of detritus. Mixing of nutrients is a positive term, adding to $N$ according to the sign of the gradient between
$N_0$ and $N$. The general direction of the nutrient flux is from a variable and nutrient-rich bottom layer to the upper layer. This
nutrient flux supports phytoplankton growth, which is the only loss term for $N$.

$$\frac{dN}{dt} = K(N_0 - N) + \beta(1 - \epsilon)(G_P + G_D) + m_D\,D - \mu_P\,P \tag{5}$$

The growth rate of phytoplankton $\mu_P$ is the product of the temperature-dependent maximum growth rate $\mu_P^{max}(T)$ and the
growth-dependencies on light ($\gamma^I$) and nutrients ($\gamma^N$), in units of $\mathrm{d}^{-1}$:

$$\mu_P = \mu_P^{max}(T)\,\gamma^I\,\gamma^N \tag{6}$$

The temperature of the upper mixed layer $T$ (in $^\circ$C) is supplied from external forcing. Under the assumption of balanced
growth, the maximum growth rate of phytoplankton $\mu_P^{max}(T)$ in $\mathrm{d}^{-1}$ is equivalent to the temperature-dependent maximum
photosynthetic rate $V_P^{max}(T)$ in $\mathrm{gC(gChl)}^{-1}\mathrm{h}^{-1}$, when converted by multiplying 24 h and considering a fixed Carbon-to-
Chlorophyll ratio of 75 $\mathrm{gC(gChl)}^{-1}$ (Sathyendranath et al., 2009). The function is parameterized via the maximum photosyn-
thetic rate at 0 $^\circ$C, represented as $V_P^{max}(0)$ ($\mathrm{gC(gchl)}^{-1}\mathrm{h}^{-1}$). The temperature dependence is calculated via the Eppley curve
(Eppley, 1972).

$$V_P^{max}(T) = V_P^{max}(0)\,1.066^T \tag{7}$$

Nutrient limitation of phytoplankton growth $\gamma^N$ is described by Michaelis-Menten kinetics:

$$\gamma^N = \frac{N}{k_N + N} \tag{8}$$





where $k_N$ (μM N) is the half-saturation constant.

The term $\gamma_I$ represents growth-dependence on light $I(z)$ available to phytoplankton through the variable depth ($z$) of the upper mixed layer. $I$ decays exponentially with $z$ (m):

$$I(z) = I_0 \, e^{(-k_{PAR} \, z)} \tag{9}$$

$I_0$ is the Photosynthetically Active Radiation (PAR), the irradiance reaching the top of the ocean surface (i.e., at $z = 0$), which is supplied from external forcing. The attenuation coefficient $k_{PAR}$ (m$^{-1}$) is the sum of light attenuation due to water, $k_w$ (0.04 m$^{-1}$), and due to the presence of phytoplankton (self-shading), accounted for by a term proportional to the concentration of phytoplankton $k_c \cdot P$ (with $k_c$ as 0.03 (μM N m)$^{-1}$), thus:

$$k_{PAR} = k_w + k_c \cdot P \tag{10}$$

We use the Smith function to calculate the photosynthetic rate (Anderson, 1993):

$$V_P = \frac{\alpha \, I(z) \, V_P^{max}}{\sqrt{(V_P^{max})^2 + \alpha^2 I(z)^2}} \tag{11}$$

Where $V_P^{max}$ is the maximum photosynthetic rate, $\alpha$ (gC(gChl)$^{-1}$h$^{-1}$(Wm$^{-2}$)$^{-1}$) is the slope of the P-I curve, and $I(z)$ is irradiance as a function of the upper mixed layer depth ($z$), Equation 9.

The light-limitation on phytoplankton growth $\gamma^I$ is then calculated by integrating $V_P$ through the upper mixed layer (i.e.,
from $z = 0$ to $z = H$).

In order to test various levels of model complexity, we also consider light attenuation according to a three-layer model of the upper mixed layer (Anderson, 1993). This alternative formulation calculates multiple $k_{PAR,i}$, with i = 1 for the top 5 m, i = 2 for the depth range 5 - 23 m and i = 3 for depths below 23 m. The changing spectral properties of water are taken into account by polynomial coefficients ($b_{0,i}$ to $b_{5,i}$).

$$k_{PAR,i} = b_{0,i} + b_{1,i}C^{1/2} + b_{2,i}C + b_{3,i}C^{3/2} + b_{4,i}C^2 + b_{5,i}C^{5/2} \tag{12}$$

where $C$ represents the chlorophyll concentration (converted as described above from μM N via $\theta_{chl}$ and the Redfield ratio). The values of the polynomial coefficients are adapted from Anderson et al. (2015) and shown in Table A1 in the appendix.

Non-grazing mortality of phytoplankton is described by the sum of linear $m_P$ (d$^{-1}$) and quadratic $m_{P2}$ ((μM N)$^{-1}$d$^{-1}$) terms (Yool and Popova, 2011). The former accounts for natural mortality and excretion. The latter describes higher order loss
processes, including for example viral infection. All non-grazing phytoplankton loss terms fuel the detritus pool.

$$\frac{dP}{dt} = \mu_P \, P - m_P \, P - m_{P2} \, P^2 - G_P - K \, P \tag{13}$$





Zooplankton graze upon phytoplankton and detritus. The grazing function is a sigmoidal (or Holling Type 3) grazing response (Anderson et al., 2015):

$$G_P = I_Z \left( \frac{\hat{\varphi}_P P}{(k_Z)^2 + \hat{\varphi}_D D + \hat{\varphi}_P P} \right) Z \tag{14}$$

where $\hat{\varphi}_P = \varphi_P \, P$ and $\hat{\varphi}_D = \varphi_D \, D$.

This formulation describes the total biomass of phytoplankton that is grazed $G_P$ (µM N). Parameter $I_Z$ (d$^{-1}$) is the maximum ingestion rate of the food source, in this case both phytoplankton and detritus. The density-dependent grazing preference parameters $\varphi_P$ and $\varphi_D$ (both dimensionless) do not represent a discrete fraction of the amount grazed in the diet relative to the environment. Instead, this amount is represented by the ratio of $\hat{\varphi}_P$ and $\hat{\varphi}_D$.

Grazing on detritus is defined as

$$G_D = I_Z \left( \frac{\hat{\varphi}_D D}{(k_Z)^2 + \hat{\varphi}_D D + \hat{\varphi}_P P} \right) Z \tag{15}$$

Zooplankton food ingestion does not directly convert into biomass. The total biomass grazed ($G_P + G_D$) is fractionated into zooplankton growth (to $Z$), excretion of dissolved nutrients (to $N$) and egestion of fecal matter & particles (to $D$). Zooplankton growth is a product of total biomass grazed ($G_P$) and the gross growth efficiency (GGE) of zooplankton. The two parameters

defining GGE in this model are absorption efficiency $\beta$ and net production efficiency $\epsilon$ (both dimensionless). Adsorption efficiency $\beta$ describes the fraction of $G_P$ that is absorbed in the gut, of which the fraction $\epsilon$ is actually assimilated into biomass (to $Z$: $\beta\epsilon$) and the rest is excreted as dissolved nutrient (to $N$: $\beta(1-\epsilon)$). GGE is the product of $\epsilon$ and $\beta$, for which values between 0.2 and 0.3 have been observed for a wide range of zooplankton (Straile, 1997). The fraction of $G_P$ egested to $D$ (e.g., as fecal pellets) is calculated via $1 - \beta$. See Anderson et al. (2015) for a more detailed discussion of this grazing formulation.

Similar to phytoplankton mortality, a linear mortality factor $m_Z$ (d$^{-1}$) represents natural mortality and excretion of zooplankton and fuels the detritus pool. A quadratic factor $m_{Z2}$ ((µM N)$^{-1}$d$^{-1}$) describes higher order predation on zooplankton, for example from fish, which is removed from the system.

$$\frac{dZ}{dt} = \beta \, \epsilon (G_P + G_D) - m_Z \, Z - m_{Z2} \, Z^2 - K \, Z \tag{16}$$

The detritus concentration in the upper layer ($D$) is fueled by mortality of phytoplankton, linear zooplankton mortality, and

zooplankton egestion (e.g., fecal pellets). The loss terms are remineralization, grazing, mixing, and additional sinking. Detritus is remineralized into $N$ at a constant rate $m_D$ (d$^{-1}$). Similar to $P$ and $Z$, a fraction of $D$ is lost due to mixing through the term $K$. In addition to $K$, a portion of detritus is lost due to gravitational sinking at a rate $v_D$ (m d$^{-1}$).

$$\frac{dD}{dt} = m_P \, P + m_{P2} \, P^2 + m_Z \, Z + (1 - \beta)(G_P + G_D) - G_D - m_D \, D - K \, D - \frac{v_D}{H} \, D \tag{17}$$





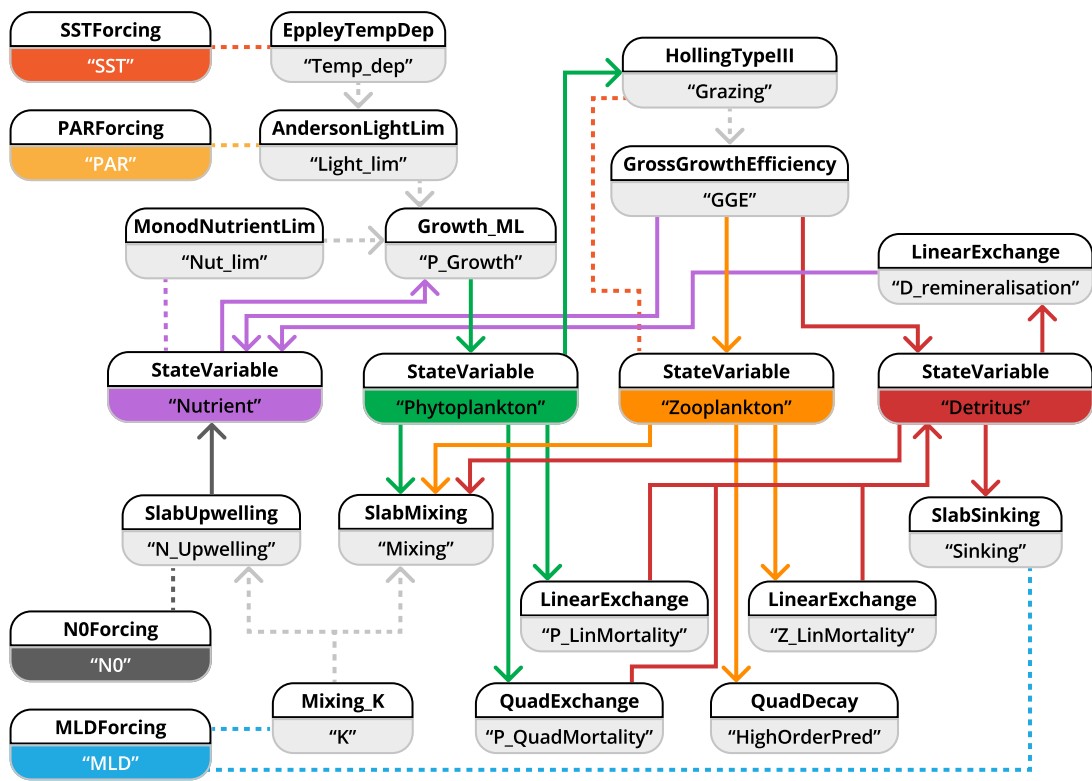

**Figure 6.** Schematic representation of how the NPZD slab-ocean model is implemented with the XSO framework and included in the Phydra library. To simplify visualization, we show only the XSO components with their labels and links. Each component contains various variables, forcings, or parameters. Solid arrows indicate the flow of fluxes between state variables. Dashed arrows indicate fluxes passed along as group variables. Dashed lines connecting processes indicate variables and forcings referenced in another component via their label.

### 3.2.2 Implementation

The ecological description of our model system is adapted from the EMPOWER model, however the technical implementation using the XSO framework is quite different from the procedural R script of Anderson et al. (2015). Instead of using hard-coded flags to choose different ecological formulations, the XSO component structure provides an object-oriented modular interface. The XSO framework defines functions irrespective of the specific time-step used for evaluation, and logically separates the model formulation from the solving algorithm in the XSO backend. This allows formulating the model without the rather complicated nested for-loop structure evaluating each time steps in the original R implementation.

The fluxes and interdependencies between the calculations in this application require a more elaborate component structure. As for the previous model application, we first separate the model into state variables, forcing, and fluxes. State variables include nutrient ($N$, Equation 5), phytoplankton ($P$, Equation 13), zooplankton ($Z$, Equation 16), and detritus ($D$, Equation





**Table 2.** Parameters considered for the NPZD model applied to four ocean stations.

| Description | Parameter | BIOTRANS | India | Papa | KERFIX | Units |
|---|---|---|---|---|---|---|
| Max. rate of photosynthesis at 0 °C | $V_P^{max}(0)$ | 2.5 | 2.5 | 1.25 | 1.25 | $gC(gChl)^{-1}h^{-1}$ |
| Initial slope of P-I curve | $\alpha$ | 0.15 | 0.15 | 0.075 | 0.075 | $gC(gChl)^{-1}h^{-1}(Wm^{-2})^{-1}$ |
| Half-saturation constant for N uptake | $k_N$ | 0.85 | 0.85 | 0.85 | 0.85 | µM N |
| Linear P mortality | $m_P$ | 0.015 | 0.015 | 0.015 | 0.015 | $d^{-1}$ |
| Quadratic P mortality | $m_{P2}$ | 0.025 | 0.025 | 0.025 | 0.025 | $(µM N)^{-1}d^{-1}$ |
| Z max. ingestion rate | $I_Z$ | 1.0 | 1.0 | 1.25 | 2.0 | $d^{-1}$ |
| Z half-saturation for intake | $k_Z$ | 0.6 | 0.6 | 0.6 | 0.6 | µM N |
| Grazing preference: P | $\varphi_P$ | 0.67 | 0.67 | 0.67 | 0.67 | dimensionless |
| Grazing preference: D | $\varphi_D$ | 0.33 | 0.33 | 0.33 | 0.33 | dimensionless |
| Z absorption efficiency | $\beta_Z$ | 0.69 | 0.69 | 0.69 | 0.69 | dimensionless |
| Z net production efficiency | $k_{NZ}$ | 0.75 | 0.75 | 0.75 | 0.75 | dimensionless |
| Linear Z mortality | $m_Z$ | 0.02 | 0.0 | 0.02 | 0.02 | $d^{-1}$ |
| Quadratic Z mortality | $m_{Z2}$ | 0.34 | 0.34 | 0.34 | 0.34 | $(µM N)^{-1}d^{-1}$ |
| D linear sinking rate | $v_D$ | 6.43 | 6.43 | 6.43 | 6.43 | $m\ d^{-1}$ |
| D remineralization rate | $m_D$ | 0.06 | 0.06 | 0.06 | 0.06 | $d^{-1}$ |
| Constant diffusive mixing | $\kappa$ | 0.13 | 0.13 | 0.13 | 0.13 | $m\ d^{-1}$ |
| Carbon-to-Chlorophyll ratio | $\theta_{chl}$ | 75 | 75 | 75 | 75 | $gC\ (gChl)^{-1}$ |

Parameters considered for the NPZD model applied to the four stations. These are optimized parameters, adapted from Anderson et al. (2015), which we employ to recreate their results. For consistency within this manuscript, we modified the mathematical symbols.

17). Forcing to the model are the upper mixed layer depth ($H$), nutrient concentration below the upper mixed layer ($N_0$),

temperature in the upper mixed layer ($T$), and irradiance at surface ($I_0$). The model defines ten unique fluxes: Phytoplankton growth, zooplankton grazing, nutrient upwelling, mixing, sinking, remineralization, and four mortality terms.

In implementing this model within the XSO framework, we aim to find a balance between component refactoring and structural simplicity. Our goal is to allow for every ecologically relevant term to be exchangeable, whilst making full use of the flexible dimensionality features. This resulted in the structure presented in Fig. 6.

To highlight one aspect of our implementation, each factor affecting phytoplankton growth is defined by an individual component. The "group to argument" feature of the XSO framework allows for such a setup to remain highly modular, since the output of each flux with the appropriate label is utilized in the product of growth limiting terms. Similarly, the component calculating the mixing coefficient $K$, is computed only once and passed along to two other components, one to calculate nutrient upwelling and the other to calculate mixing loss fluxes of phytoplankton, zooplankton, and detritus. A user could

readily add more growth limiting terms via new components or exchange the component calculating $K$ without necessitating any changes to the rest of the model or workflow.




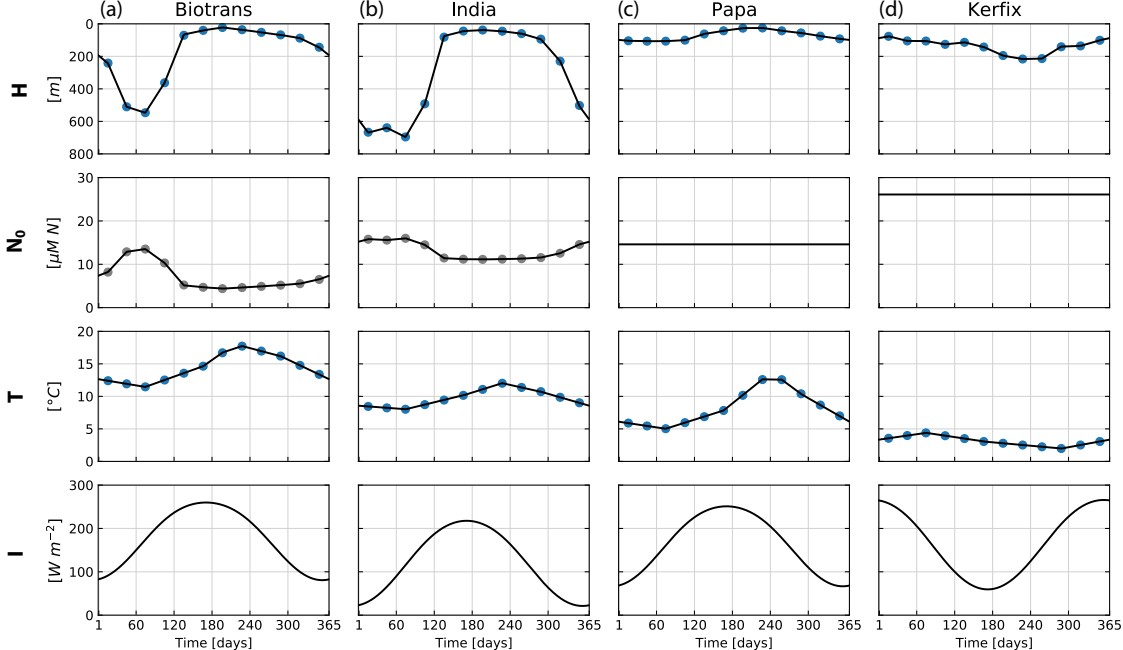

**Figure 7.** Forcing corresponding to the four locations considered for the NPZD model application. Mixed Layer Depth ($H$), Nitrate below the Mixed Layer ($N_0$), irradiance at surface ($I$), and temperature averaged through the upper mixed layer ($T$). Forcing data is calculated from the IFREMER MLD climatology and WOA 2018 data. The mixed layer depth (MLD) is extracted from a global MLD climatology (de Boyer Montégut, 2022) and used to extract the temperature of the mixed layer from WOA 2018 climatology data (Garcia et al., 2019). The blue dots indicate data extracted from monthly climatology, the gray dots are calculated values from this data. Nutrient forcing $N_0$ is a function of depth for locations Biotrans and India and a constant value for Papa and Kerfix. Irradiance is calculated as a function of latitude, following Anderson et al. (2015).

Following Anderson et al. (2015), we compare model performance in four locations representing named ocean stations: BIOTRANS, India, Papa, and KERFIX. We present the parameters in Table 2, which were optimized for the specific locations by Anderson et al. Two of the stations are located in the temperate North Atlantic, BIOTRANS (47°N, 20°W) and India (60°N, 335 20°W), both of which exhibit a characteristic phytoplankton spring bloom, followed by a phase of low nutrient availability during summer. The other two stations, Papa in the North Pacific (50°N, 145°W) and KERFIX in the Southern Ocean (50°40'S, 68°25'E), represent High-Nutrient-Low-Chlorophyll (HNLC) environments with a much less pronounced seasonal cycle. The contrasting environments are clearly discernible from the forcing data (see Fig. 7) In each location, the NPZD slab model is forced by the four corresponding environmental factors. The forcing for the Mixed Layer Depth ($H$) is taken from an updated 340 version of the IFREMER MLD climatology (de Boyer Montégut et al., 2004), calculated using a fixed density threshold criterion of $0.03 \, \mathrm{kg}^{-1}\mathrm{m}^3$ from 10 m depth value (de Boyer Montégut, 2022). The nutrient concentration below the mixed layer ($N_0$) is calculated from a combination of the MLD climatology and depth-resolved climatology for nitrate in the World Ocean





Atlas (WOA) 2018 (Garcia et al., 2019). The temperature of the mixed layer ($T$) was calculated using the MLD climatology and the temperature data of WOA 2018 (Locarnini et al., 2019). The monthly climatological data are interpolated to match

the number of model time steps. Anderson et al. (2015) used a liner interpolation and, for comparability, we adopted the same approach. The forcing for irradiance at surface ($I_0$) is calculated via a light submodel that employs trigonometric and astronomical equations to calculate light at a given location, with latitude and cloud fraction as input parameters (for exact formulation, please see Appendix A, Anderson et al. (2015)).

To highlight another technical aspect, we use the batch dimension feature of the XSO model setup function to evaluate the

model for all four stations in unison. This feature allows us to define a new dimension at model setup and to supply a list of values for parameters of that dimension. In our case, this additional dimension defines the four stations via the specific forcing and the parameters $V_P^{max}$, $\alpha$, $I_Z$ and $m_Z$, which are location-specific (see Table 2). At runtime, the model is solved for each set of parameters in the supplied lists and outputs are returned in a single Xarray dataset. The model outputs for each station can be easily retrieved via the supplied batch dimension label. This feature is also very useful for exploring parameter ranges

(e.g., for sensitivity analysis).

We additionally show a modification of the model: Anderson et al. (2015) included a detailed discussion of the treatment of light in a slab model. From the formulations presented in the original paper, we consider two implementations. These are the simple Beer's law, which parameterizes light attenuation with a single attenuation coefficient for the whole upper mixed layer (see equation 10), and the more elaborate piecewise description, which evaluates light attenuation in three discrete depth

intervals within the upper mixed layer, with specific polynomial coefficients for each interval (see equation 12). Model results for both formulations are presented in the following section.

### 3.2.3    Results

Model outputs for the four stations are shown in Fig. 8. Following Anderson et al. (2015), the output of the state variables $N$ and $P$ are compared to climatological data from the locations. For $N$, the model output is compared to the concentration of

nitrate within the upper mixed layer, that is calculated from a combination of WOA 2018 nitrate data (Garcia et al., 2019) and IFREMER MLD Climatology (de Boyer Montégut et al., 2004). Phytoplankton concentration ($P$) is compared to converted chlorophyll data extracted for the locations from MODIS Aqua climatology retrieved up until August 2022 (NASA Goddard Space Flight Center, 2018). In order to simplify the presentation, all units are given as concentration of Nitrogen μM N. The chlorophyll concentration data is converted by a constant factor $\theta_{chl}$ (75 gC (gChl)$^{-1}$) and the Redfield ratio of 6.625

molC(molN)$^{-1}$ as assumed C:N of phytoplankton. We use climatology data, because we do not assume to be able to replicate particular biomass peaks of certain years with climatological forcing. The climatological data follows the general pattern shown in the chlorophyll data used as verification data in the original paper, which was taken from a specific representative year.

The climatological data shows a marked seasonal cycle visible with a clear spring phytoplankton bloom for stations BIO-TRANS and India, as expected, given their location in the temperate North Atlantic. Stations Papa and KERFIX show less

pronounced cycles, but still some seasonal variation, with generally higher phytoplankton and zooplankton concentrations in summer (in their respective hemisphere). Zooplankton and detritus dynamics clearly follow phytoplankton concentrations, as



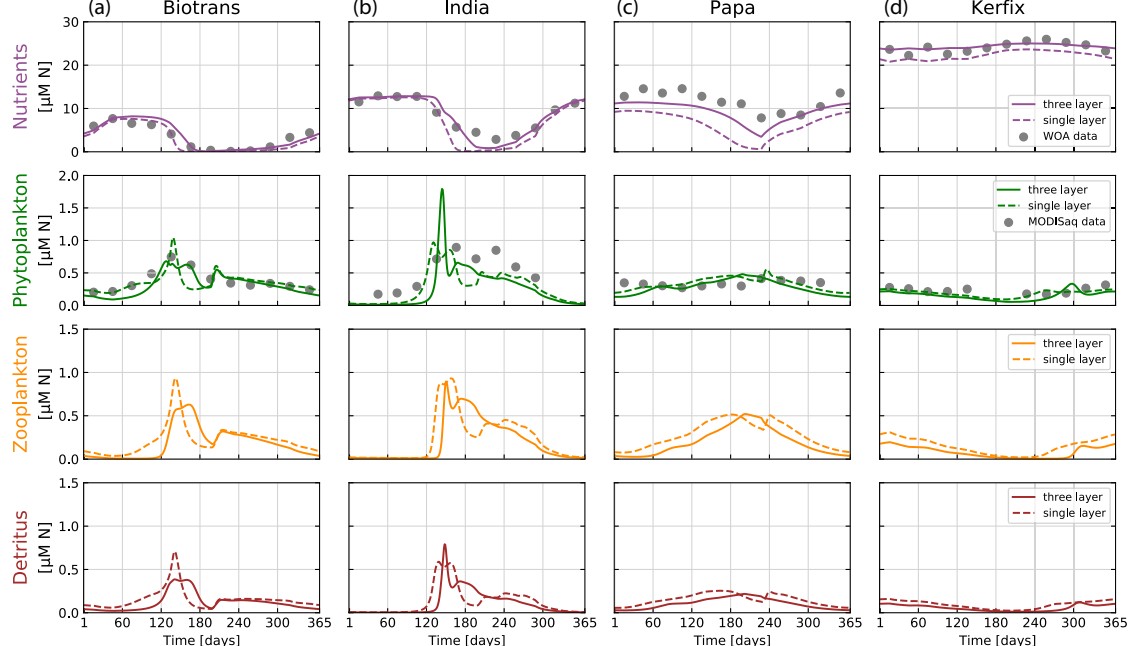

**Figure 8.** Results of the NPZD model (application 2) for locations (a) BIOTRANS, (b) India, (c) Papa, and (d) KERFIX. We show here the final year of a five-year run, allowing for model spin-up. Model output is shown for two model variants in relation to the light attenuation algorithm used, with everything else being kept equal (see parameters in Table 2). The dashed lines show model outputs using the simple Beer's law for light attenuation (calculated over the entire mixed layer). The solid lines are outputs from the model variant that resolves light attenuation over three discrete depth layers. The data for nitrogen in the upper mixed layer (grey dots) are extracted from WOA 2018, using IFREMER MLD climatology. Phytoplankton nitrogen biomass (grey dots) is calculated via $\theta_{chl}$ and Redfield ratios from MODIS Aqua chlorophyll monthly climatologies for the specific locations. For some months, no satellite data is available for stations Papa, India and KERFIX.

expected. In general, the model output agrees relatively well with our verification data, with the optimized parameters from Anderson et al. (2015). In accordance with their results, the change in light attenuation treatment has a pronounced effect on nutrient dynamics, as well as some effect on phytoplankton growth. The model results obtained with light attenuated according to the three-layer formulation show a better agreement with the data, particularly for station Papa. Nutrient draw-down during growth periods is consistently lower when compared to the simple Beer's law. This is caused by a greater effect of phytoplankton concentration on the resulting $k_{PAR}$ (light attenuation factor).

These results show that our framework can recreate accurately the results of published marine ecosystem modelling studies within a flexible and modular environment, which allows further experimentation and testing of different model structures.





## 3.3 Model application 3: size-based Nutrient-Phytoplankton-Zooplankton (NPZ) model

Our third model application is a size-structured plankton community model in an idealized physical setting, similar to a chemostat. The presented model is an adaptation of the ASTroCAT model, developed by Neil Banas (Banas, 2011). ASTroCAT was developed as a tool to investigate complex trophic interactions between phytoplankton and zooplankton in a simplified setting, resolving a diverse plankton community via a size spectrum. Cell or organism size is used in this model as a "master trait", defining the parameters of specific plankton types via allometric functions, taken from literature (Litchman and Klausmeier, 2008). This allows for a functional and quantifiable model to investigate mechanisms affecting and sustaining phytoplankton diversity.

Banas considered model dynamics under variable forcing or with stochastic grazing parameters. Here, we focus on the basic parameter setup under constant forcing. While trophic interactions between phytoplankton and zooplankton size classes are highly resolved, other ecological processes are neglected (e.g., there are no detrital or regeneration pathways).

This model lends itself well to highlight the flexibility of the XSO framework. A state variable defined within a *component* can be defined with a dimension label, so that it can represent an array of state variables of flexible size, as long as dimension labels match across *components* in the same model. The size of the state variable array depends on the number of values supplied at model setup. The built-in vectorization allows the model to compute correctly and efficiently, even with large numbers of state variables. We showcase this feature by running the model with 2 to 50 size classes and comparing bulk phytoplankton biomass between runs. The only modification necessary is varying the number of values supplied at model setup.

### 3.3.1 Description

The model expresses quantities in units of μM N. The physical setting is analogous to a chemostat with constant nutrient inflow counterbalanced by permanent losses. There is no explicit outflow process implemented, but mortality and egestion fluxes are simply lost from the system.

The model describes size-structured communities of phytoplankton and zooplankton, whose sizes are expressed in terms of Equivalent Spherical Diameter (ESD). Following Banas (2011), we run our initial simulations with 40 size classes of equally log-spaced $P$ (1 to 20 μm), and 40 size classes of $Z$ (2.1 to 460 μm). Additionally, we perform an experiment in which the number of size classes within these ranges is varied from 2 to 50. The model can be defined with any number of size classes within meaningful boundaries of allometric relationships. Size classes are denoted by the subscript $i$ for phytoplankton and $j$ for zooplankton.



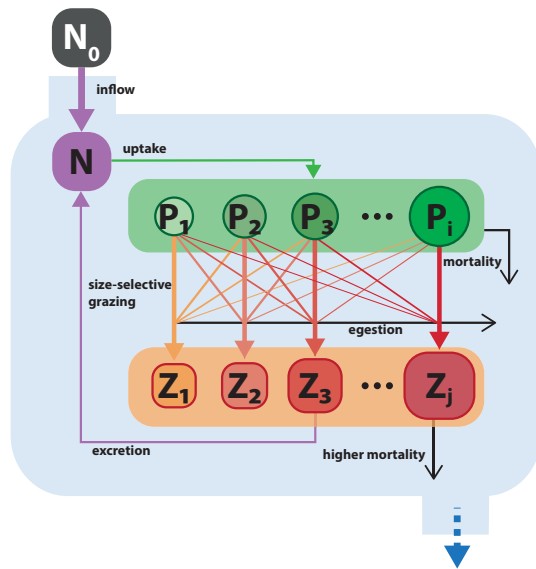

**Figure 9.** Schematic of the size-resolved $NP_iZ_j$ trophic model. Model structure and parameterization are adapted from Banas (2011). Boxes with black and white labels represent, respectively, state variables and external forcing. Arrows indicate fluxes between state variables. The blue boundary contains the ecosystem model, with state variables for a nutrient and multiple size classes of phytoplankton and zooplankton. Filled colored arrows represent exchanges between state variables, open black arrows represent fluxes that are lost from the model system.

Model nutrient $N$ (µM N) is resupplied from an external source, with concentration $N_0$ (µM N) and delivered at a constant rate $f$ (d$^{-1}$). In addition, a fraction of grazed biomass that is not assimilated by $Z$ (units) is returned to the nutrient pool. The only loss term for $N$ is phytoplankton nutrient uptake.


$$\frac{dN}{dt} = f\,N_0 + (1 - \epsilon - f_{eg}) \sum_j \sum_i G_P^{ij} - \sum_i (\mu_{max}^i \, \gamma_i^N \, P_i) \tag{18}$$

Each phytoplankton size class $P_i$ (µM N) grows according to Michaelis-Menten kinetics:

$$\gamma_i^N = \frac{N}{k_N^i + N} \tag{19}$$

where $\gamma_i^N$ is the limitation on phytoplankton growth due to nutrients, $k_N^i$ (µM N) is the size-dependent half saturation

constant, and $N$ is the ambient nutrient concentration.





Phytoplankton loss due to natural mortality and excretion is described with the factor $m^P$ (units) that is scaled by the maximum intrinsic growth rate $\mu_{max}^i$ (d$^{-1}$), so that $m^P \mu_{max}^i$ yields the specific mortality rate for each size class.

$$\frac{dP_i}{dt} = \mu_{max}^i \, \gamma_i^N \, P_i - m_P \, \mu_{max}^i \, P_i - \sum_j G_P^{ij} \tag{20}$$

The grazing of the zooplankton size class $Z_j$ (µM N) on the phytoplankton size class $P_i$ is calculated by

$$G_P^{ij} = \mu_j^Z \, \frac{\varphi_{ij} \cdot P_i}{k_Z + \sum_i (\varphi_{ij} \cdot P_i)} \, Z_j \tag{21}$$

where $I_Z^j$ (d$^{-1}$) is the size-dependent maximum ingestion rate, $k_Z$ (µM N) is the half-saturation constant and $\varphi_{ij}$ (dimensionless) is the relative preference of $Z_j$ for $P_i$.

Prey preference is assumed to vary with phytoplankton size $size_P^i$ (µm) in a log-normal distribution around an optimal prey size for each grazer $size_{opt}^j$ (µm).

$$\varphi_{ij} = exp\left[-\left(\frac{log_{10}(size_P^i) - log_{10}(size_{opt}^j)}{\Delta size_P}\right)\right] \tag{22}$$

Where $\Delta size_P$ is the prey size tolerance parameter ($log_{10}$(µm)ESD) that controls the width of the Gaussian distribution.

Zooplankton growth is calculated as the product between total biomass grazed ($G_P$) and gross growth efficiency ($\epsilon$), for which values between 0.2 and 0.3 have been observed for a wide range of zooplankton (Straile, 1997). A fraction $f_{eg}$ of grazed biomass is assumed to be quickly excreted to $N$ and another fraction ($\epsilon$) that would feed into a detrital pool is permanently lost
from the system. Following Banas (2011), the grazing fractions are split equally so that $\epsilon = f_{eg} = 1/3$.

Zooplankton experience quadratic losses according to the parameter $m_{Z2}$, scaled by the total sum of $Z_j$. This term describes higher-order mortality and predation on zooplankton and is permanently removed from the system.

$$\frac{dZ_j}{dt} = \epsilon \sum_i G_P^{ij} - m_{Z2} \, Z_j \sum_j Z_j \tag{23}$$

### 3.3.2 Implementation

Parameters were adapted from Banas (2011), see Table 3 for all used parameter values and allometric relationships.

We separate the model into state variables, forcing, and fluxes. State variables are nutrient ($N$, Equation 18), multiple size-classes of phytoplankton ($P_i$, Equation 20), and multiple size-classes of zooplankton ($Z_j$, Equation 23). The only forcing is the external nutrient input ($N_0$). At least 5 fluxes (of variable dimensionality based on the number of zooplankton and phytoplankton) can be defined: The inflow of the external medium, $P_i$ growing on $N$, $Z_j$ grazing on $P_i$, and mortality terms
for $P_i$ and $Z_j$. The model is implemented using 10 XSO components (Fig. 10). We simplify the schematic by only showing the components with their respective labels.

The original ASTroCAT model was implemented with an interactive graphical user interface showing animations of model outputs. Our implementation in the XSO framework lacks this, but provides some technical updates, with the major differences



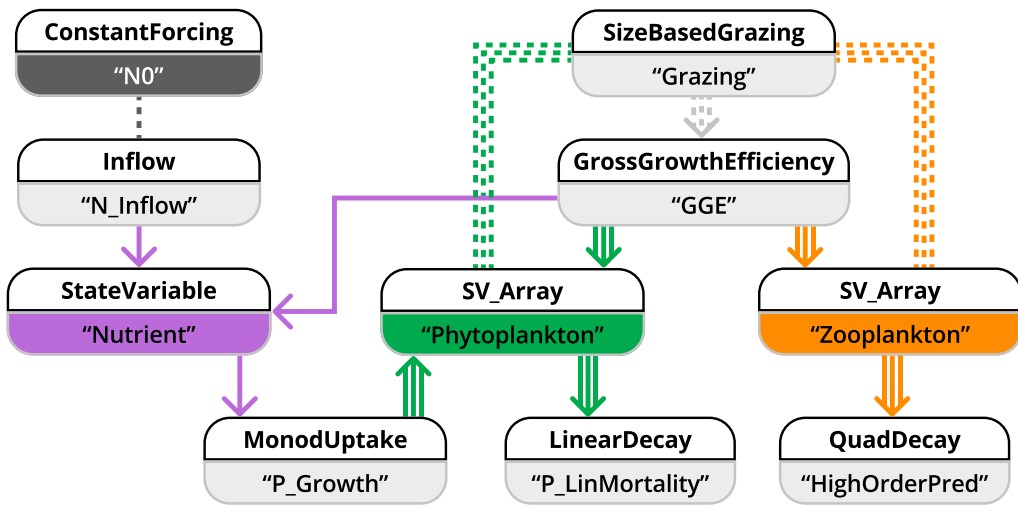

**Figure 10.** Schematic representation of how model application 3 is implemented in the XSO framework and included in the Phydra library. For simplicity, only the XSO components with corresponding labels and links are shown. Each component consists of a number of variables, forcing, or parameters. Solid arrows indicate the fluxes between state variables. Dashed arrows indicate fluxes passed along as group variables. Dashed lines connecting processes indicate variables and forcing passed along via their label. Arrows with multiple lines indicate values with dimensions that are passed along.

being the modular component structure and the use of vectorization (instead of for-loops) to define functions computing the
*fluxes* acting on arrays of size-classes.

Banas (2011) presented a detailed analysis of model output for variable metrics of ecosystem complexity. We recreated only one part of the original analyses, with a simple comparison of model dynamics for a variable number of phytoplankton and zooplankton size classes. The number of state variables can be varied at model setup by supplying a list of initial values with the desired dimensions. We ran the model for the range of 2 to 50 size classes.

### 3.3.3   Results

Running the model with 40 size classes of phytoplankton and zooplankton recreates the dynamics originally presented by Banas (2011). See Fig. 11 for the time evolution of $N$, $P_i$ and $Z_j$ over a ten-year run. The size-resolved food-web shows oscillatory changes in biomass with periods from days to years, despite the much faster growth rates in the model. There appear to be trade-offs between size classes, driven by the selective grazing interactions between zooplankton and phytoplankton. This,
however, does not lead to chaotic behavior, but instead tends towards a stable state after 5 years of model run. Interestingly, the general dynamics, as well as the stable state is highly clustered into some size classes. As Banas (2011) discussed, this





**Table 3.** Parameters and allometric functions used for the size-based NPZ model.

| Description | Parameter | Value | Units |
|---|---|---|---|
| Flow rate of external nutrient | $f$ | 1 | $\mathrm{d}^{-1}$ |
| External nutrient concentration | $N_0$ | 1 | µM N |
| Prey half-saturation constant [3] | $k_Z$ | 3 | µM N |
| Prey size tolerance [4] | $\Delta size_P$ | 0.25 | $\log_{10}$ µm |
| Mortality fraction of $\mu_{max}^i$ for $P_i$ | $m_P$ | 0.1 | $\mathrm{d}^{-1}$ |
| Zooplankton growth efficiency | $\epsilon$ | 0.33 | dimensionless |
| Fraction of grazing egested | $f_{eg}$ | 0.33 | dimensionless |
| | | | |
| Maximum growth rate of $P_i$ [1] | $\mu_{max}^i$ | $2.6\,\mathrm{d}^{-1}\left(\frac{size_P^i}{1\,\mu\mathrm{m}}\right)^{-0.45}$ | $\mathrm{d}^{-1}$ |
| Nutrient half-saturation constant of $P_i$ [2] | $k_N^i$ | $0.1\,\mu\mathrm{M\,N}\left(\frac{size_P^i}{1\,\mu\mathrm{m}}\right)$ | µM N |
| Maximum ingestion rate of $Z_j$ [3] | $I_Z^j$ | $26\,\mathrm{d}^{-1}\left(\frac{size_Z^j}{1\,\mu\mathrm{m}}\right)^{-0.4}$ | $\mathrm{d}^{-1}$ |
| Optimum prey size of $Z_j$ [4] | $size_{opt}^j$ | $0.65\,\mu\mathrm{m}\left(\frac{size_P^i}{1\,\mu\mathrm{m}}\right)^{0.56}$ | µm |

Parameters adapted form Banas (2011). Original sources: [1] Tang (1995), [2] Eppley et al. (1969), [3] Hansen et al. (1997), [4] Hansen et al. (1994)

"banding" seems to be a direct result of the prey preferences. A general conclusion one can draw is that selective grazing interactions can be a strong factor in structuring plankton communities.

To investigate the effect of the number of resolved size classes on the model output, we conduct comparative model runs varying the number of phytoplankton and zooplankton between 2 and 50. Fig. 12 shows the effect on bulk phytoplankton biomass when running the model with a variable number of size classes. A lower number of size classes (2-10) show highly variable outputs. Bulk dynamics seems to stabilize for numbers of size classes above 10. However, there are still deviations between runs in relation to the average phytoplankton biomass when more than 10 size classes are considered. The increased size resolution seems to reduce the perturbations dependent on initial model conditions, confirming the patterns observed by Baird and Suthers (2010).

## 4 Discussion

We argue that codes of plankton community models are often built to be run, but not to be shared, reused and modified, which is in part an issue related to the programming languages and tools used to create them. This is in contrast to current computational tools for data analysis (for example, as developed by the Python or R programming communities), that focus on modularity, usability, and clear documentation, in an open-source, collaborative context.

The XSO framework in its current version allows building models quickly and dynamically from *components* and provides a user interface to setup and run a model that is stored as a fully documented Xarray dataset. The Phydra library provides a set




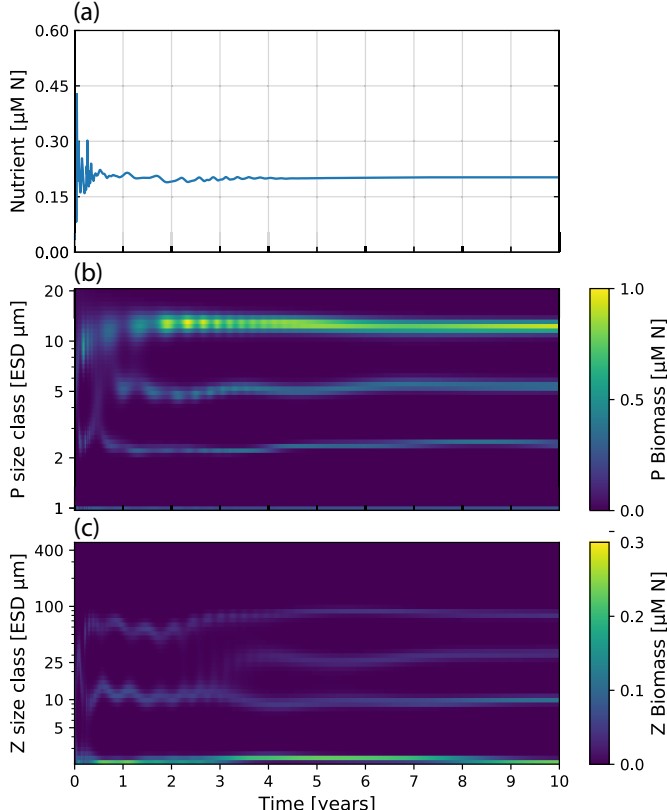

**Figure 11.** Nutrient concentration and plankton biomass under steady nutrient forcing obtained with model runs resolving 40 phytoplankton and zooplankton size classes. Size classes are log-spaced in the range of 1 to 20 μm for phytoplankton and 2.16 to 420 μm for zooplankton. (a) Nutrient concentration over time. (b) Phytoplankton biomass by size class over 10 years of model time evolution. (c) Zooplankton biomass over the same period.

of *components*, models, and example applications that showcase the usability of the framework and provide a common library for marine ecosystem modelling applications. The first release of the Phydra library, presented here, contains implementations
of two published plankton ecosystem models, the EMPOWER model by Anderson et al. (2015) and the ASTroCAT model by Banas (2011).

## 4.1 Structuring complex marine ecosystem models in a flexible framework

There has been an increasing move towards developing and using frameworks that systematize or simplify at least one specific aspect of model development (e.g., FABM for model coupling, Bruggeman and Bolding (2014)). However, their usage is quite
scattered in the scientific community (Janssen et al., 2015). The design choices of a modeling framework have a profound effect





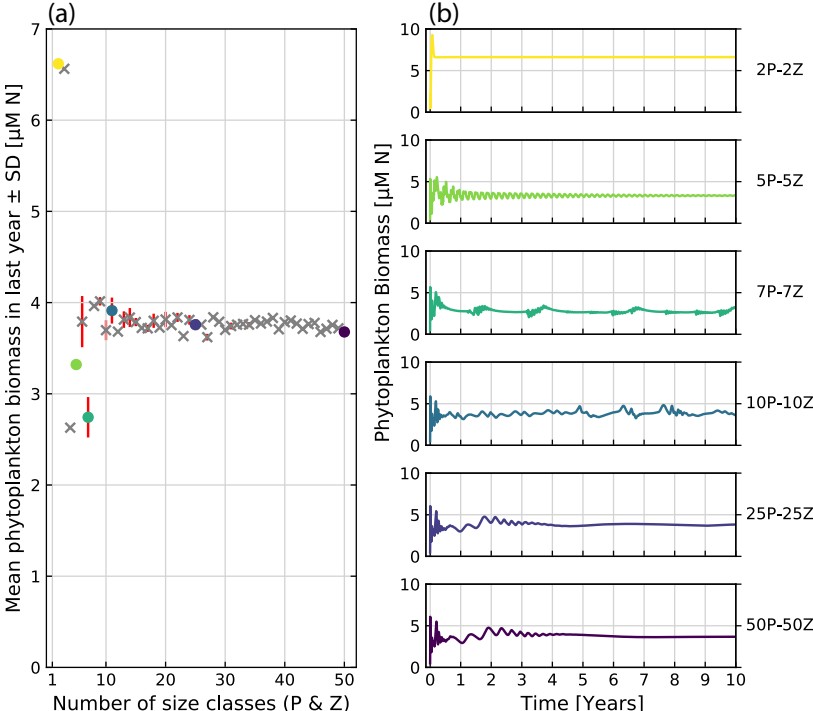

**Figure 12.** Comparative runs of our implementation of the ASTroCAT model with varying numbers of size-classes of Phytoplankton and Zooplankton. (a) Mean biomass of phytoplankton in the last year of a ten-year run for a range of 2 to 50 size classes of phytoplankton and zooplankton. Standard deviation is plotted in red. Grey crosses mark runs not otherwise shown, colored dots correspond to exemplary runs. (b) Exemplary model runs. The sum of phytoplankton biomass is shown over a ten-year run.

on both the flexibility and usability, with an inherent trade-off between these two aspects. In developing the Phydra library, we went through many iterations, with the logical conclusion being the separation of the framework and library aspects.

Our goal in developing the framework was to allow users to build models without restricting the level of complexity, in particular in relation to the dimensionality, number of state variables and model processes. This was implemented in the framework by providing *variable types*, which directly correspond to the basic mathematical components of models based on ordinary differential equations (e.g., state variables, parameters, forcing, and partial equations). Every aspect of the model needs to be defined at the level of *variable types*. Model *components* can be flexibly constructed from the provided set of *variable types* and wrap a logical component of the model as users see fit. State variables, forcing and parameters need to be initialized in one *component*, but can be referenced across the model. The system of differential equations is constructed from the *fluxes* contained in the model *components* via the supplied labels at model setup. These design choices make the effort required to construct models proportional to the desired model complexity, and *components* can be easily modified to more complex formulations. In order to provide a template for utilizing this flexible framework, we present fully implemented





models in the Phydra library. We hope that this will foster experimentation and inter-comparison of model performance at different levels of complexity.

In addition to flexible model construction, we wanted to provide an interface for iterative modification and prototyping. An ecosystem model tracks chemical compounds and ecosystem components via state variables. These state variables can define completely different components of a model, or represent functional groups. In the third model application, we presented such a case by defining an array of variables for phytoplankton and zooplankton via size-based allometric functions. This flexible dimensionality of model components was designed with the current issues in marine ecosystem modeling in mind. The effects

of different levels of complexity in the number and definition of phytoplankton functional types (PFT), for example, is not routinely tested in marine ecosystem models (Franks, 2009). Phydra provides a framework that allows for easy testing through flexible modification of such model complexity at model setup.

The choice of programming language has an important effect on the resulting framework. In contrast to available tools that allow building models based on differential equations from a set of customizable building blocks through a graphical interface

(e.g., Stella, PowerSim, Ecopath) or other frameworks that utilize a custom scripting language (e.g., via YAML files), the Phydra and XSO frontend and backend are fully implemented in a single programming language: Python. This might require a higher initial effort for users unfamiliar to Python, but we argue that the effort is worth given the wealth of functionality provided by the Python scientific ecosystem and the support of the large community of programmers and developers. The XSO model development workflow is similar to writing standard Python codes, with the added benefit of having at hand a set of

modular Python objects and attributes that automatically handle model inputs and outputs and that allow to computationally construct and run models.

Since the XSO framework is fully implemented in Python, functional model *components* have to follow a basic structure, but are otherwise flexible. The functions defining *forcings* and *fluxes* within *components* in XSO are not restrictive in their Python syntax and can make use of external Python packages, as long as the value that is finally supplied at model runtime is compati-

ble with the chosen solver backend. Since XSO itself is a wrapper of Xarray-simlab without hiding its underlying functionality, XSO further expands the possibilities for custom applications and further development of the Xarray-simlab framework. The relative complexity of the backend framework should not dissuade users less interested in technical customization, as the Phydra library provides fully functional pre-configured *components* and *model objects* that provide a blueprint for the development of marine ecosystem models using XSO.

The software presented here was specifically designed to support collaborative model development. Scientists working with computational models do not always build the models themselves. Often, scientists use existing models and focus the work on parameterization and analysis of results obtained with model applications in specific locations. This type of use is specifically supported in our software because we equipped the Phydra library with pre-built *model objects* and *components*. A user can start working with models without detailed knowledge of the underlying framework and learn the basic workflow before

progressing to building custom models using the XSO framework. Additionally, more advanced users can easily share custom *components* or *model objects* via the respective Python objects. This particular feature of design makes our software suitable also for teaching.





## 4.2 Current limitations of XSO and Phydra

The presented software packages are in the early stages of development, and as such have limited functionality. This first
version of the XSO framework supports mathematical models based on ordinary differential equations. In the first release, the
framework functionality and library contents are focused on zero-dimensional physical settings for marine plankton models.
The first version of XSO implements two numerical solvers. These are (1) a simple step-wise solver and (2) an adaptive step-
size solver optimized for solving a system of ODEs (`solve_ivp` from the SciPy package). The simple step-wise solver is
the only backend that currently supports multi-model parallelism when executing multiple sets of parameters via the *batch*
dimensionality feature. None of the implemented solvers currently support single model parallelism and are thus not optimized
for very large models (i.e., more than 200 state variables). There are also limits to the flexibility of the framework, particularly
for reusing components between models, since the dimensionality of a flux or state variable is hard-coded in the component
and can not be altered after creating a *model object*.

## 4.3 Current usage and future developments

XSO is available via the package installer for Python (pip). Detailed instructions about installation and resolution of depen-
dencies can be found in the online documentation (Post, 2023b). Since Python and the dependencies of Phydra are constantly
developed, we provide instructions there on how to install a fully compatible virtual environment with the Conda manager sep-
arated from a user's standard Python installation (Post, 2023a). For interactive coding and prototyping of models using Phydra,
we recommend using the Jupyter notebook environment that is available via Conda. For more complex and larger model runs
on servers or clusters, Python scripts are preferable.

The Xarray-simlab package that provides the basis for the XSO framework is a relatively young project, but has found robust
usage in, for example, the Fastscape package (Bovy, 2021) which is continuously developed and used. Since XSO provides a
flexible wrapper around Xarray-simlab, and the XSO solver backend is implemented in an adaptable object-oriented manner,
further developments can proceed without necessarily impacting already implemented models.

Since the XSO framework is embedded in the larger Python scientific ecosystem, there are many possibilities to provide ad-
vanced functionality on top of the basic model development workflow currently supported. Amongst our foremost development
goals are developing the solving backend further to support larger models and possibly multi-dimensional physical settings.
The solver backend could be adapted to use highly optimized solvers, such as the Mobius framework (Norling et al., 2021).
Another important aspect would be simplifying the process of parameter optimization and sensitivity analysis. We are also
working on methods for model introspection, such as graphically representing the model structure and exporting the system of
equations.

The Phydra library of *components* and *model objects* could be expanded beyond the three applications presented here and
would allow easy comparability and reproducibility of specific model applications, as demonstrated here.



## 5 Conclusions

We presented two new Python packages that provide a flexible tool-set for plankton community models based on differential equations. Phydra is a library that offers a library of pre-built models their individual building blocks (i.e., *components*), which can be combined or modified to create custom configurations. The XSO package, which is the technical foundation of Phydra, provides a user interface and modeling framework for building and solving computational models based on differential equations. The XSO framework grants users granular control over state variables, parameters, forcing, and mathematical functions, while allowing each model component to remain interchangeable. Additionally, Phydra utilizes the Xarray dataset format for structuring model input and output, including metadata, allowing for easy storage, sharing, and analysis of data. The Phydra library in the initial release contains three model applications of variable ecosystem complexity, from a simple chemostat model to a size-resolved plankton model. These three applications are contained in the Phydra library via their respective model *components* and as fully assembled *model objects*. Additionally, all scripts used to create the presented results are available in fully documented Jupyter notebooks.

The Phydra library can be a reference and learning resource for scientists interested in marine ecosystem modelling, a starting point for scientific exploration, and a valuable tool for teaching. The model development effort is proportional to the desired complexity of the model application, so users can quickly implement simple models. Further developing such a fully integrated environment for marine ecosystem modeling will require a diverse community of users and developers. We believe the programming language Python provides strong enough functionalities and a wide enough user base. Hence, Phydra and XSO can contribute to the ongoing efforts of developing more robust, transparent, and reproducible models, moving away from monolithic and inflexible codes to a model development process that is inherently collaborative.

*Code availability.* Xarray-simlab-ODE (XSO) and Phydra are fully open source and available under a BSD-3 license on GitHub. The XSO framework is available via Post (2023b), DOI: https://doi.org/10.5281/zenodo.8178616, and the Phydra library is available via Post (2023a) DOI: https://doi.org/10.5281/zenodo.8178694

*Author contributions.* BP conceived of and wrote the Phydra and XSO Python packages. BP, EAT, AB and AM conceived of the manuscript structure and model applictions presented. BP wrote the draft, with EAT, AB and AM contributing to revisions.

*Competing interests.* The contact author has declared that none of the authors has any competing interests.

*Acknowledgements.* We want to thank the open-source developers, that made this work possible. In particular, we want to thank Benoît Bovy for advice on some aspects of the code structure, and for developing Xarray-simlab.



**Table A1.** Coefficients for use in the three-layer light attenuation model for the NPZD model application

| First layer (0-5 m) | Second layer (5-23 m) | Third layer (>23 m) |
|---|---|---|
| $b_{0,1} = 0.13096$ | $b_{0,2} = 0.041025$ | $b_{0,3} = 0.021517$ |
| $b_{1,1} = 0.030969$ | $b_{1,2} = 0.036211$ | $b_{1,3} = 0.050150$ |
| $b_{2,1} = 0.042644$ | $b_{2,2} = 0.062297$ | $b_{2,3} = 0.058900$ |
| $b_{3,1} = -0.013738$ | $b_{3,2} = -0.030098$ | $b_{3,3} = -0.040539$ |
| $b_{4,1} = 0.0024617$ | $b_{4,2} = 0.0062597$ | $b_{4,3} = 0.0087586$ |
| $b_{5,1} = -0.00018059$ | $b_{5,2} = -0.00051944$ | $b_{5,3} = -0.00049476$ |

Originally presented in Anderson (1993).

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
