# Peer review of "The XSO framework (v0.1) and Phydra library (v0.1) for a flexible, reproducible and integrated plankton community modeling environment in Python"

_EGUsphere, 2023_

## Author Comment (AC1)

Reviewer comments are blue and *italicized,* Replies in normal font.

**Reviewer #1**

We thank the anonymous reviewer for the thoughtful and constructive comments.

**1.1 Expanding dimensionality and coupling to circulation models**

*In the current state the XSO framework only supports zero-dimensional models and I fear that this will be a significant concern for many potential users. It is necessary to expand on the single sentence in line 536 that addresses this issue. What is the future potential for expanding on the dimensionality and coupling to physical circulation? In the introduction (line 30-32) the authors mention the problem of linking ecological "legacy" models to physical models but without the possibility to couple or use a physical circulation with the XSO framework many users will need to have several versions of the code (XSO and Fortran) which is time consuming.*

It is correct that, in the current version, the XSO framework does not contain tools to facilitate multi-dimensional physical settings, but there is no limit in the dimensionality of variables, fluxes and parameters. There is also no technical limitation in the XSO framework that may prevent the implementation of multi-dimensional physical settings or coupling to circulation models. In addition, we had mentioned in the Discussion section that actively supporting multi-dimensional physical settings is among our next development goals.

With the current release (version 0.1), the model building blocks can be implemented with an optional set of higher dimensions (dims=['time', ('z','time'), ('lat','z','time'), …]), which specifically allows for this type of flexibility. What is currently missing are the helper functions and an optimized solver backend, that allow a user to readily set up multi-dimensional physical settings. This substantial amount of work  goes beyond the scope of  the current release. We have opened a corresponding GitHub Issue on our repository (https://github.com/ben1post/xarray-simlab-ode/issues/1), to create a channel for discussing with interested users the developments in this direction.

According to the design principles of XSO, multi-dimensional models should be implemented in a straightforward and modular manner. In principle, any zero-dimensional model built with the XSO framework should be transferable to a 1D, 2D or 3D setting using

provided functions or components. This can take the form of a set of custom components that define the interaction between grid-points, in addition to helper functions, that facilitate the creation of multi-dimensional model input variables and parameters.

Ideally, these future developments should be compatible with the FABM framework, in order to enable users to couple a biogeochemical model developed with XSO to the hydrodynamic models embedded in FABM. Technically, this would be possible, as there exists a wide range of functionality in Python to interact with other programming languages such as Fortran, and FABM already provides a Python API via the "pyfabm" package. However, the considerable amount of work required by these objectives is out of the scope of the current manuscript.

We will add these explanations in the revised manuscript.

**1.2 Computational efficiency**

*In model application 2 and 3 (line 315 and 449) the authors note that the XSO framework allows for vectorization of nested for-loops. However, the authors do not address the computational efficiency of the new framework. Is it possible to compare the model implications in XSO to the original models? The issue of computational efficiency is on the front of every develops mind when they choose a framework for model development.*

This is a very valuable point. As a benchmark, we ran a simple timed execution of the original EMPOWER model implementation written in R versus our implementation of the model using the XSO Framework in Python. The original R implementation took on average 3.84 seconds to execute a five-year run and store the output results in variables on a modern Macbook Pro (16 GB RAM, M1 Pro processor). The XSO version took 0.396 seconds on average for the same tasks.

Various novel features of our XSO framework may explain the difference, including: (1) the built-in vectorization of model functions, which is more efficient than evaluating "for loops", (2) the use of an optimized adaptive step-size solver in the form of "solve_ivp" RK45 from the SciPy library, and (3) the modular component structure not requiring additional "if-else" statements to define a variant of the model. Another relevant feature to note is that adding a "batch" dimension to the input parameters (see our EMPOWER implementation), and providing the parameter "parallel=True" at runtime, allows multiple model runs to be executed in parallel. This allows taking advantage of multiprocessor computers to considerably speed up parameter scans and sensitivity tests.

We will add these points in the Discussion section of the revised manuscript.

**1.3 ODE integration method**

*The only method for integration is RK45. However, plankton models are commonly stiff, which requires a solver that is designed for stiff systems. Could another integration method be added? Alternatively, this concern should be flagged in the discussion and suggested as a future development.*

Thank you for pointing this out. The only presented integration solver was RK45, but the XSO solver backend implements the "solve_ivp" module provided by the SciPy package in Python, and does allow all solvers contained in that module to be used. We limited the first release to RK45, but we have already updated the package to provide direct access to the LSODA and BDF solvers, which are more suitable to stiff systems. These features will be available with the next release of XSO.

We will mention these solvers in the Discussion section of the revised manuscript.

**1.4 Phydra as a codebase for scientific exploration**

*At line 147-150 the authors propose the idea that Phydra could be a codebase for scientific exploration. It is not, however, clear how that should be accomplished. Do the authors want others to add their models directly to their GitHub? Without further specification it is not realistic that this will become this peer-reviewed database the authors hope it will.*

The framework (XSO) and the library (Phydra) are separate, therefore it is possible for other authors to contribute to the library. Taking advantage of hosting our project open-source on GitHub, users can raise issues with specific requests that we can try to address. Or, alternatively, the potential contributors can develop and integrate their model by cloning the Phydra repository and then submitting merge requests. We are happy to check for compatibility and help users in the implementation of their models into Phydra.

We will add a sentence on the procedure for contributors in the revised manuscript and in the documentation of the library in GitHub.

**1.5 Version number**

It seems appropriate to change the version number from 0.1 to 1.0.

Thank you for the suggestion. We denote our first release as "0.1.0" following the semantic versioning system typically used in software development. We prefer to keep the 0.1 version for this release because the current version is considered an initial development release, and not yet a full version. Some important aspects, such as the treatment of dimensionality and other user-facing functions are still under development. Nevertheless, we deem this release complete and useful for the scientific community and its publication now is crucial for inviting collaboration and discussion.

**Minor technical/editorial issues:**

*Title: The title mentions "reproducible" models. However, all models that publish their code are reproducible – even though the codes are often messy and inflexible (this is also an issue in line 5 of the abstract).*

It is true that all published codes are reproducible, but we meant the term not exclusively, but as "more so". XSO and Phydra were designed to specifically allow all stages of model development to be directly documented and easily shared.

XSO allows labeling variables and parameters directly in the code (e.g., with the appropriate units) and this information is also contained as metadata in the structured model output Xarray datasets. Model development and analysis can be performed in Juypter Notebooks, that allow documenting the code with multiple media and text types in an interactive environment, that can run in any modern web browser. The notebooks contained in the Phydra library, provide a proof of concept and template for such a "highly reproducible" model development workflow. We will add some clarification of this term to the Introduction and Abstract of the revised manuscript.

*Title: The title mention "integrated". It is unclear what that refers to.*

Here, "integrated" refers to the modeling environment in a technical sense. Model, data and analysis are embedded seamlessly in the same computational environment. The term was taken from Laniak et al. 2013 ("Integrated environmental modeling: a vision and roadmap for the future."). Every part of the modeling process can be performed in the Python scientific ecosystem. The modeling framework XSO is directly integrated with the data structure of Xarray, which is in turn integrated with packages such as Pandas and Matplotlib for data analysis and visualization. We will add some clarification of this term to the Introduction and Abstract of the revised manuscript.

*Line 18. This list of references are largely irrelevant to the statement. They are examples of models, but not a review of the history of modelling. I suggest to remove the references, as they are not really needed either.*

For the revised manuscript, we have now removed all unnecessary references, except for Gentleman, 2022 ("A chronology of plankton dynamics in silico: how computer models have been used to study marine ecosystems"), which pertains to the history of phytoplankton modeling.

*Line 19: "comprising" => "comprised"*

We will correct this in the revised manuscript

*Line 21. There is something odd going on with the sentence here. Please reformulate.*

Agreed, we will reformulate this sentence to make it more clear.

*Line 66. A more appropriate reference with be Evans & Parslows whose "slab" model is essentially a chemostat (one section of their paper is with a fixed MLD, which is exactly a chemostat).*

Thank you for the suggestion, we will add Evan & Parslows as a reference.

*Equ. 14. This is not a type III, but a type II functional relationship, with a half staturation constant $k_Z^2$.*

Thank you. We will correct this in the revised manuscript.

*Line 430. Mathematical functions (exp, log) should not be written in italics*

We will correct this in the revised manuscript.

**Reviewer #2**

We thank the anonymous reviewer for the positive and constructive feedback.

**2.1 Only 0-D and no ability to integrate with physical models**

*While the introduced environment holds great promise as a valuable tool for teaching plankton modeling, it may still require further development before it can be effectively utilized for scientific investigations. This limitation stems from its current sole reliance on 0D physical settings or its inability to integrate with physical models. The authors have acknowledged the necessity of coupling with other hydrodynamic models and have proposed the inclusion of multi-dimensional physical settings in the next development. However, it remains unclear how these enhancements would be practically achieved within the XSO framework, especially considering the author's mention of limitations related to the hard-coded dimensionality of flux or state variables in the components and can not be altered after creating a model object. It would be beneficial if the authors could provide more details here.*

Thank you for raising this issue. Please also see our response to a similar question from Reviewer #1 above. The modular solver backend and existing functionalities for flexible dimensionality provide a clear "entry point" for further developing these capabilities. It is possible to provide an optional list of higher dimensions for variables, which would be suitable to support variable physical settings even after creating a model object.

We will add these points and those mentioned above to the Discussion of the revised manuscript and we will clarify our roadmap towards implementing multi-dimensional physical settings in the Phydra library.

**2.2 Learning curve and GUI**

*From Phydra on GitHub (Post, 2023a), it seems that users might be required to invest time in learning the parameters, variables, and functions of the Phydra library and XSO framework in Python. This learning curve, when compared to other frameworks (e.g., FABM), might not offer a significant advantage. However, if Phydra and XSO were to be provided through a Graphical User Interface (GUI), this could significantly enhance their usability and accessibility.*

A GUI with a drag and drop interface to build components from variable types and models from components would indeed be possible to implement. This goes beyond the scope of

the current release, but is on our roadmap for future development.

The XSO framework already provides a level of abstraction, similar to a GUI, for users building model components from variable types. Using the XSO variable types and components removes the need to allocate variables and implement a solver, and instead allows a user to focus on the model formulations and parameterization. The straightforward compact Python syntax should allow users to start building custom models faster than when working with existing Fortran code, and the resulting model is much easier to modify and analyze. The modular and extensible nature of XSO allows advanced users to develop highly efficient and flexible model development workflows.

Note also that FABM and Phydra do not have the same scope. In contrast to FABM, Phydra is focused on developing biogeochemical models, and could be further developed to actually utilize FABM for coupling such models to hydrodynamic models. In fact, FABM provides a Python API that could be used for such purposes.

We will clarify these aspects in the revised version of the manuscript.

**2.3 Comparison to FABM**

*In the broader context of ecosystem modeling, enhancing flexibility and accessibility through frameworks like XSO and Phydra is imperative. Given the existence of other frameworks, such as FABM, which offers a wide array of hydrodynamic and biogeochemical models, it would be valuable to understand the authors' vision for XSO and Phydra within the ecosystem modeling community. Specifically, what role or contribution do the authors envision for XSO and Phydra, and how do they intend to develop and position these tools within this field?*

As mentioned in the previous point, FABM and our framework have a different scope. FABM is a framework for the interface between biogeochemical models and hydrodynamic models. With further development, the XSO framework can provide the toolbox to construct biogeochemical models in a highly interactive and modular manner, and FABM could provide the robust linking to physical models. As of now, XSO and Phydra present an attempt to bring the ease of use and flexibility of modern data-analysis workflows in Python to the ecosystem modeling community, for Phydra the focus is on enabling highly flexible plankton community models.

These objectives will require a continued developmental effort, and we would welcome any contribution from interested modelers via GitHub Issues or Pull Requests. To further clarify

the difference to, and potential synergies with FABM, we will add these points to the Discussion section of the revised manuscript.

---

## Author Response (AR1)

Reviewer comments are blue and *italicized,* Replies in normal font. Line number references the LatexDiff version of the revised manuscript (with tracked changes).

**Reviewer #1**

We thank the anonymous reviewer for the thoughtful and constructive comments.

**1.1 Expanding dimensionality and coupling to circulation models**

*In the current state the XSO framework only supports zero-dimensional models and I fear that this will be a significant concern for many potential users. It is necessary to expand on the single sentence in line 536 that addresses this issue. What is the future potential for expanding on the dimensionality and coupling to physical circulation? In the introduction (line 30-32) the authors mention the problem of linking ecological "legacy" models to physical models but without the possibility to couple or use a physical circulation with the XSO framework many users will need to have several versions of the code (XSO and Fortran) which is time consuming.*

**Response:** It is correct that, in the current version, the XSO framework does not contain tools to facilitate multi-dimensional physical settings, but there is no limit in the dimensionality of variables, fluxes and parameters. There is also no technical limitation in the XSO framework that may prevent the implementation of multi-dimensional physical settings or coupling to circulation models. In addition, we had mentioned in the Discussion section that actively supporting multi-dimensional physical settings is among our next development goals.

With the current release (version 0.1), the model building blocks can be implemented with an optional set of higher dimensions (dims=['time', ('z','time'), ('lat','z','time'), ...]), which specifically allows for this type of flexibility. What is currently missing are the helper functions and an optimized solver backend, that allow a user to readily set up multi-dimensional physical settings. This substantial amount of work  goes beyond the scope of  the current release. We have opened a corresponding GitHub Issue on our repository (https://github.com/ben1post/xarray-simlab-ode/issues/1), to create a channel for discussing with interested users the developments in this direction.

According to the design principles of XSO, multi-dimensional models should be implemented in a straightforward and modular manner. In principle, any zero-dimensional model built with the XSO framework should be transferable to a 1D, 2D or 3D setting using provided functions or components. This can take the form of a set of custom components that define the interaction between grid-points, in addition to helper functions, that facilitate the creation of multi-dimensional model input variables and parameters.

Ideally, these future developments should be compatible with the FABM framework, in order to enable users to couple a biogeochemical model developed with XSO to the hydrodynamic models embedded in FABM. Technically, this would be possible, as there exists a wide range of functionality in Python to interact with other programming languages such as Fortran, and FABM already provides a Python API via the "pyfabm" package. However, the considerable amount of work required by these objectives is out of the scope of the current manuscript.

**Changes to manuscript:** We have added explanations in the following sections of the revised manuscript: Discussion of current limitations in Section 4.2 line 568 - 575, p. 29. Further discussion of future developments in Section 4.3 in line 604 - 616, p. 30.

**1.2 Computational efficiency**

*In model application 2 and 3 (line 315 and 449) the authors note that the XSO framework allows for vectorization of nested for-loops. However, the authors do not address the computational efficiency of the new framework. Is it possible to compare the model implications in XSO to the original models? The issue of computational efficiency is on the front of every develops mind when they choose a framework for model development.*

**Response:** This is a very valuable point. As a benchmark, we ran a simple timed execution of the original EMPOWER model implementation written in R versus our implementation of the model using the XSO Framework in Python. The original R implementation took on average 3.84 seconds to execute a five-year run and store the output results in variables on a modern Macbook Pro (16 GB RAM, M1 Pro processor). The XSO version took 0.396 seconds on average for the same tasks.

Various novel features of our XSO framework may explain the difference, including: (1) the built-in vectorization of model functions, which is more efficient than evaluating "for loops", (2) the use of an optimized adaptive step-size solver in the form of "solve_ivp" RK45 from the SciPy library, and (3) the modular component structure not requiring additional "if-else" statements to define a variant of the model. Another relevant feature to note is that adding

a "batch" dimension to the input parameters (see our EMPOWER implementation), and providing the parameter "parallel=True" at runtime, allows multiple model runs to be executed in parallel. This allows taking advantage of multiprocessor computers to considerably speed up parameter scans and sensitivity tests.

**Changes to manuscript:** We have added this information as a paragraph to the revised manuscript in the Discussion in Section 4.1 starting at line 553, p. 28.

**1.3 ODE integration method**

*The only method for integration is RK45. However, plankton models are commonly stiff, which requires a solver that is designed for stiff systems. Could another integration method be added? Alternatively, this concern should be flagged in the discussion and suggested as a future development.*

**Response:** Thank you for pointing this out. The only presented integration solver was RK45, but the XSO solver backend implements the "solve_ivp" module provided by the SciPy package in Python, and does allow all solvers contained in that module to be used. We made use of the RK45, but two different backward differentiation algorithm solvers more suitable for stiff systems are also available (specifically these options are called "LSODA" and "BDF").

**Changes to manuscript:** We have added a mention of the other solver options at line 135, p. 6, and some more explanation to the Discussion in Section 4.2 starting at line 577, p. 29.

**1.4 Phydra as a codebase for scientific exploration**

*At line 147-150 the authors propose the idea that Phydra could be a codebase for scientific exploration. It is not, however, clear how that should be accomplished. Do the authors want others to add their models directly to their GitHub? Without further specification it is not realistic that this will become this peer-reviewed database the authors hope it will.*

**Response:** The framework (XSO) and the library (Phydra) are separate, therefore it is possible for other authors to contribute to the library. Taking advantage of hosting our project open-source on GitHub, users can raise issues with specific requests that we can try to address. Or, alternatively, the potential contributors can develop and integrate their model by cloning the Phydra repository and then submitting merge requests. We are happy to check for compatibility and help users in the implementation of their models into Phydra.

**Changes to manuscript:** We have expanded on the existing paragraph in Section 2.2 at line 161, p. 7.

**1.5 Version number**

It seems appropriate to change the version number from 0.1 to 1.0.

**Response:** Thank you for the suggestion. We denote our first release as "0.1.0" following the semantic versioning system typically used in software development. We prefer to keep the 0.1 version for this release because the current version is considered an initial development release, and not yet a full version. Some important aspects, such as the treatment of dimensionality and other user-facing functions are still under development. Nevertheless, we deem this release complete and useful for the scientific community and its publication now is crucial for inviting collaboration and discussion.

**Minor technical/editorial issues:**

*Title: The title mentions "reproducible" models. However, all models that publish their code are reproducible – even though the codes are often messy and inflexible (this is also an issue in line 5 of the abstract).*

**Response:** It is true that all published codes are reproducible, but we meant the term not exclusively, but as "more so". XSO and Phydra were designed to specifically allow all stages of model development to be directly documented and easily shared.

XSO allows labeling variables and parameters directly in the code (e.g., with the appropriate units) and this information is also contained as metadata in the structured model output Xarray datasets. Model development and analysis can be performed in Juypter Notebooks, that allow documenting the code with multiple media and text types in an interactive environment, that can run in any modern web browser. The notebooks contained in the Phydra library, provide a proof of concept and template for such a "highly reproducible" model development workflow.

**Changes to manuscript:** We have added some clarification of this term to the revised manuscript. Specifically, in the Abstract at line 10 and 12, p.1, and in Section 2.1 at line 127 - 131, p. 6.

*Title: The title mention "integrated". It is unclear what that refers to.*

**Response:** Here, "integrated" refers to the modeling environment in a technical sense. Model, data and analysis are embedded seamlessly in the same computational environment. The term was taken from Laniak et al. 2013 ("Integrated environmental modeling: a vision and roadmap for the future."). Every part of the modeling process can be performed in the Python scientific ecosystem. The modeling framework XSO is directly integrated with the data structure of Xarray, which is in turn integrated with packages such as Pandas and Matplotlib for data analysis and visualization.

**Changes to manuscript:** We have added some clarification of this term to the revised manuscript. Specifically, in the Abstract at line 11, p.1, and in the last sentence of the Figure 1 caption on p. 4.

*Line 18. This list of references are largely irrelevant to the statement. They are examples of models, but not a review of the history of modelling. I suggest to remove the references, as they are not really needed either.*

**Changes to manuscript:** At line 22, p.2, we have now removed all unnecessary references, except for Gentleman, 2022 ("A chronology of plankton dynamics in silico: how computer models have been used to study marine ecosystems"), which pertains to the history of phytoplankton modeling.

*Line 19: "comprising" => "comprised"*

**Changes to manuscript:** Corrected at line 22, p.2.

*Line 21. There is something odd going on with the sentence here. Please reformulate.*

**Changes to manuscript:** The sentence was reformulated, see line 23 - 25, p.2.

*Line 66. A more appropriate reference with be Evans & Parslows whose "slab" model is essentially a chemostat (one section of their paper is with a fixed MLD, which is exactly a chemostat).*

**Changes to manuscript:** Thank you for the suggestion, we have added Evan & Parslows as a reference at line 181, p. 7.

*Equ. 14. This is not a type III, but a type II functional relationship, with a half staturation constant $k_Z^2$.*

**Response:** After rechecking the formulation and the original publication from which the formulation was directly adapted (Anderson et al. 2015), we suppose it is indeed a type III (sigmoidal) grazing function. Please let us know if we are mistaken.

**Changes to manuscript:** We have more clearly formulated that the equation was directly adapted from Anderson et al. (2015) at line 297, p. 14.

*Line 430. Mathematical functions (exp, log) should not be written in italics*

**Changes to manuscript:** We have corrected this in Equation 22 in the revised manuscript at line 445 on p. 22.

**Reviewer #2**

We thank the anonymous reviewer for the positive and constructive feedback.

**2.1  Only  0-D  and  no  ability  to  integrate  with physical models**

*While the introduced environment holds great promise as a valuable tool for teaching plankton modeling, it may still require further development before it can be effectively utilized for scientific investigations. This limitation stems from its current sole reliance on 0D physical settings or its inability to integrate with physical models. The authors have acknowledged the necessity of coupling with other hydrodynamic models and have proposed the inclusion of multi-dimensional physical settings in the next development. However, it remains unclear how these enhancements would be practically achieved within the XSO framework, especially considering the author's mention of limitations related to the hard-coded dimensionality of flux or state variables in the components and can not be altered after creating a model object. It would be beneficial if the authors could provide more details here.*

**Response:** Thank you for raising this issue. Please also see our response to a similar question from Reviewer #1 above. The modular solver backend and existing functionalities for flexible dimensionality provide a clear "entry point" for further developing these capabilities. It is possible to provide an optional list of higher dimensions for variables, which would be suitable to support variable physical settings even after creating a model object.

**Changes to manuscript:** We have added explanations in the following sections of the revised manuscript: Discussion of current limitations in Section 4.2 line 568 - 575, p. 29. Further discussion of future developments in Section 4.3 in line 604 - 616, p. 30.

**2.2 Learning curve and GUI**

*From Phydra on GitHub (Post, 2023a), it seems that users might be required to invest time in learning the parameters, variables, and functions of the Phydra library and XSO framework in Python. This learning curve, when compared to other frameworks (e.g., FABM), might not offer a significant advantage. However, if Phydra and XSO were to be provided through a Graphical User Interface (GUI), this could significantly enhance their usability and accessibility.*

**Response:** A GUI with a drag and drop interface to build components from variable types and models from components would indeed be possible to implement. This goes beyond the scope of the current release, but is on our roadmap for future development.

The XSO framework already provides a level of abstraction, similar to a GUI, for users building model components from variable types. Using the XSO variable types and components removes the need to allocate variables and implement a solver, and instead allows a user to focus on the model formulations and parameterization. The straightforward compact Python syntax should allow users to start building custom models faster than when working with existing Fortran code, and the resulting model is much easier to modify and analyze. The modular and extensible nature of XSO allows advanced users to develop highly efficient and flexible model development workflows.

Note also that FABM and Phydra do not have the same scope. In contrast to FABM, Phydra is focused on developing biogeochemical models, and could be further developed to actually utilize FABM for coupling such models to hydrodynamic models. In fact, FABM provides a Python API that could be used for such purposes.

**Changes to manuscript:** In the introduction, the paragraph at line 127 - 131, p.6, we highlighted the specific strong points of our framework. We have added two sentences to clearly state the aim of this first release of XSO and Phydra in Section 4.1 at line 505 - 508, p. 27. We have also added differences to FABM and possible future developments in Section 4.3 at line 612, p. 30.

**2.3 Comparison to FABM**

*In the broader context of ecosystem modeling, enhancing flexibility and accessibility through frameworks like XSO and Phydra is imperative. Given the existence of other frameworks, such as FABM, which offers a wide array of hydrodynamic and biogeochemical models, it would be valuable to understand the authors' vision for XSO and Phydra within the ecosystem modeling community. Specifically, what role or contribution do the authors envision for XSO and Phydra, and how do they intend to develop and position these tools within this field?*

**Response:** As mentioned in the previous point, FABM and our framework have a different scope. FABM is a framework for the interface between biogeochemical models and hydrodynamic models. With further development, the XSO framework can provide the toolbox to construct biogeochemical models in a highly interactive and modular manner, and FABM could provide the robust linking to physical models. As of now, XSO and Phydra

present an attempt to bring the ease of use and flexibility of modern data-analysis workflows in Python to the ecosystem modeling community, for Phydra the focus is on enabling highly flexible plankton community models.

These objectives will require a continued developmental effort, and we would welcome any contribution from interested modelers via GitHub Issues or Pull Requests.

**Changes to manuscript:** To further clarify the difference to, and potential synergies with FABM, we have expanded on the description of FABM in Section 4.1 at line 499, p. 27, and highlighted the aim of this first release of XSO and Phydra at line 505 - 508, p. 27. Possible future developments were further expanded in Section 4.3 at line 612, p. 30.